# Online Tensor Max-Norm Regularization via Stochastic Optimization

**Tong Wu** *wutong@bigai.ai*
*Beijing Institute for General Artificial Intelligence*

**Reviewed on OpenReview:** *https://openreview.net/forum?id=1iDpP3GWmS*

## Abstract

The advent of ubiquitous multidimensional arrays poses unique challenges for low-rank modeling of tensor data due to higher-order relationships, gross noise, and large dimensions of the tensor. In this paper, we consider online low-rank estimation of tensor data where the multidimensional data are revealed sequentially. Induced by the recently proposed tensor-tensor product (t-product), we rigorously deduce the tensor max-norm and formulate the tensor max-norm into an equivalent tensor factorization form, where the factors consist of a tensor basis component and a coefficient one. With this formulation, we develop an online max-norm regularized tensor decomposition (OMRTD) method by alternatively optimizing over the basis component and the coefficient tensor. The algorithm is scalable to the large-scale setting and the sequence of the solutions produced by OM-RTD converges to a stationary point of the expected loss function asymptotically. Further, we extend OMRTD for tensor completion. Numerical experiments demonstrate encouraging results for the effectiveness and robustness of our algorithm. The code is available at `https://github.com/twugithub/2024-TMLR-OMRTD`.

## 1 Introduction

In the last decade or so, we have witnessed an explosion in data generation due to the development of new, affordable consumer electronics and advances in data storage and communication technologies. Many information processing tasks involve data samples that are naturally structured as multidimensional arrays, also known as tensors. Examples of tensor data include images, videos, hyperspectral images, tomographic images and multichannel electroencephalography data. Low-rank tensor estimation has attracted increasing attention in the research community owing to its successful applications within computer vision (Zhang et al., 2014), data mining (Franz et al., 2009), and signal processing (Sidiropoulos et al., 2017). In this work, specifically, we consider an observed three-way data tensor $\mathcal{Z}$, and we attempt to learn a low-rank tensor $\mathcal{X}$ that best approximates grossly corrupted observations. This problem typically involves minimizing a weighted combination of the approximation error and a penalty for the tensor rank.

This problem is well studied in the matrix domain, where the goal is to optimize the rank of the prediction matrix and this problem is likely to be computationally infeasible. The nuclear norm (Recht et al., 2010) and the max-norm (Srebro et al., 2004) are the two commonly used convex relaxations of the rank function (NP-hard). However, different from matrices, there are many tensor rank definitions because a tensor can be factorized in many ways. Common tensor decompositions include CANDECOMP/PARAFAC (CP) decomposition (Kolda & Bader, 2009), Tucker decomposition (Tucker, 1966), HOSVD decomposition (Lathauwer et al., 2000), tensor-train decomposition (Oseledets, 2011). All these decompositions consider low-rank structure in the original domain. However, there is an increasing realization that exploitation of low-rank structure in the frequency domain can lead to improved performance of many tasks (Lu et al., 2019; Song et al., 2020). In particular, a new tensor decomposition method called tensor singular value decomposition (t-SVD) (Kilmer et al., 2013; 2021) has demonstrated that after conducting Discrete Fourier Transform (DFT) along the 3rd mode, a 3-way tensor can exhibit strong low-rankness in the Fourier domain.

Several works for tensor robust principal component analysis (RPCA) relying on these decompositions are proposed (Anandkumar et al., 2016; Yang et al., 2020; Lu et al., 2020; Gao et al., 2021). While these approaches have been incredibly successful in many applications, an important shortcoming is that they are not scalable to large-scale tensor data because the memory requirements increase rapidly with the size of data. Moreover, such methods are all implemented in a batch manner, which cannot efficiently capture the dynamics of streaming data.

Motivated by the t-product (Kilmer & Martin, 2011), we rigorously deduce a new tensor max-norm for tensor decomposition. By utilizing the tensor factorization form of the proposed max-norm, we develop an efficient algorithm, termed *online max-norm regularized tensor decomposition* (OMRTD), to solve tensor max-norm regularized problem. OMRTD processes only one sample per time instance, making it scalable to large-scale tensor data. We also extend OMRTD for the tensor completion problem, where low-rank tensor data recovery is carried out in the presence of missing data. Extensive experimental results on the tensor subspace recovery task illustrate that the proposed tensor max-norm always performs better than the tensor nuclear norm in dealing with a large fraction of corruption. Effectiveness of the proposed algorithm is evaluated through online background subtraction.

The rest of this paper is organized as follows. Section 2 briefly discusses related work and Section 3 introduces some mathematical notations and tensor basics. Section 4 presents the proposed OMRTD method in the presence of complete and missing data. We present experimental results in Section 5 and provide concluding remarks in Section 6.

## 2    Related Work

Low-rank models find applications in collaborative filtering (Srebro et al., 2004), hyperspectral image restoration (He et al., 2016), and background subtraction (Candès et al., 2011). To handle data contaminated by gross corruption, the matrix RPCA (Candès et al., 2011) decomposes the observed matrix into a low-rank component and a sparse component using nuclear norm regularization. The work in Srebro et al. (2004) considered collaborative prediction and learned low-max-norm matrix factorizations by solving a semi-definite program. To establish the connection between the max-norm and the nuclear norm, Srebro & Shraibman (2005) considered collaborative filtering as an example and proved that the max-norm variant enjoys a lower generalization bound than the nuclear norm. In the large-scale setting, there are some efforts that attempted to develop efficient algorithms to solve max-norm regularized or constrained problems (Rennie & Srebro, 2005; Lee et al., 2010; Fang et al., 2018). Yet, the applicability of such batch optimization methods is problematic because of their high memory cost. To alleviate this issue, online learning approaches that are based on nuclear norm and max-norm matrix decomposition using stochastic optimization have been proposed in Feng et al. (2013) and Shen et al. (2017), respectively. However, all these approaches are devised for 2-way data, thus limiting their abilities to exploit the intrinsic structure of tensors.

Besides the tensor max-norm used in this paper, there exist several different tensor rank definitions due to the complex multilinear structure of tensors. The CP rank (Kiers, 2000) is defined as the smallest number of rank one tensor decomposition. However, the CP rank and its convex relaxation are NP-hard to compute (Hillar & Lim, 2013). To alleviate this issue, the tractable Tucker rank (Tucker, 1966) is more flexible because it explores the low-rank structure in all modes. The sum-of-nuclear-norms (SNN) is defined as the sum of the nuclear norms of unfolding matrices (Liu et al., 2013), which is served as a convex surrogate for the Tucker rank. The effectiveness of this idea has been well studied in Goldfarb & Qin (2014) and Huang et al. (2014). But it was proved in Romera-Paredes & Pontil (2013) that SNN is not the tightest convex relaxation of the Tucker rank. Recently, based on the tensor-tensor product and tensor singular value decomposition (t-SVD) scheme (Kilmer et al., 2013), a new tensor rank called tensor tubal rank (Kilmer et al., 2013) is defined as the number of nonzero singular tubes of the singular value tensor in t-SVD. Correspondingly, a new tensor nuclear norm is proposed and applied in tensor completion (Zhang & Aeron, 2017), tensor robust PCA (Lu et al., 2020), and tensor data clustering (Zhou et al., 2021; Wu, 2023). As a tensorial generalization of $k$-support norm (Argyriou et al., 2012), the $*_L$-spectral $k$-support norm is proposed in Wang et al. (2021) to exploit low-rankness in the spectral domain and has been applied for tensor recovery.

Nonetheless, all these methods require memorizing all the samples in each iteration and they cannot process samples in a sequential way. To address this concern, several online methods have been developed for streaming tensor data analysis (Yu et al., 2015; Mardani et al., 2015; Zhang et al., 2016; Kasai, 2019; Li et al., 2019; Gilman et al., 2022). Among these works, Mardani et al. (2015) and Kasai (2019) obtain multi-way decompositions of low-rank tensors with missing entries using the CP decomposition, whereas both Zhang et al. (2016) and Gilman et al. (2022) rely on the t-SVD framework. Note that Zhang et al. (2016) implements online tensor robust PCA using tensor nuclear-norm regularization, while our work adopts our proposed tensor max-norm for tensor decomposition. Indeed, since the tensor max-norm is a more complicated mathematical entity, the development of online methods for the max-norm regularization requires more attention. Based on the tensor factorization form of the proposed tensor max-norm, we convert the problem into a constrained tensor factorization problem that is amenable to online implementation.

## 3 Notations and Preliminaries

In this section, we introduce notations and some basic facts about t-SVD that will be used throughout this paper. More related tensor facts can be found in Kilmer et al. (2013; 2021). Throughout this paper, we use lowercase, bold lowercase, bold uppercase, and bold calligraphic letters for scalars, vectors, matrices, and tensors, respectively. For a three-way tensor $\boldsymbol{\mathcal{A}} \in \mathbb{R}^{n_1 \times n_2 \times n_3}$, its $(i, j, k)$-th entry is denoted as $\boldsymbol{\mathcal{A}}_{i,j,k}$. We use Matlab notation $\boldsymbol{\mathcal{A}}(i, :, :)$, $\boldsymbol{\mathcal{A}}(:, i, :)$ and $\boldsymbol{\mathcal{A}}(:, :, i)$ or $\mathbf{A}^{(i)}$ to denote the $i$-th horizontal, lateral and frontal slices, respectively. Any lateral slice of size $n_1 \times 1 \times n_3$ is denoted as $\overrightarrow{\boldsymbol{\mathcal{A}}}$. In particular, we also use $\overrightarrow{\boldsymbol{\mathcal{A}}}_i$ to denote the $i$-th lateral slice of $\boldsymbol{\mathcal{A}}$. The $(i, j)$-th mode-3 fiber is denoted by $\boldsymbol{\mathcal{A}}(i, j, :)$. The transpose $\boldsymbol{\mathcal{A}}^T \in \mathbb{R}^{n_2 \times n_1 \times n_3}$ is obtained by transposing each frontal slice of $\boldsymbol{\mathcal{A}}$ and then reversing the order of the transposed frontal slices 2 through $n_3$. We use $\bar{\boldsymbol{\mathcal{A}}} = \texttt{fft}(\boldsymbol{\mathcal{A}}, [\,], 3)$ to denote the Discrete Fourier transform along mode-3 of $\boldsymbol{\mathcal{A}}$. Similarly, $\boldsymbol{\mathcal{A}}$ can be computed from $\bar{\boldsymbol{\mathcal{A}}}$ using $\boldsymbol{\mathcal{A}} = \texttt{ifft}(\bar{\boldsymbol{\mathcal{A}}}, [\,], 3)$. The inner product between two tensors $\boldsymbol{\mathcal{A}}$ and $\boldsymbol{\mathcal{B}}$ in $\mathbb{R}^{n_1 \times n_2 \times n_3}$ is defined as $\langle \boldsymbol{\mathcal{A}}, \boldsymbol{\mathcal{B}} \rangle = \sum_{i,j,k} \boldsymbol{\mathcal{A}}_{i,j,k} \boldsymbol{\mathcal{B}}_{i,j,k}$. The $\ell_1$ and Frobenius norms of $\boldsymbol{\mathcal{A}}$ are defined as $\|\boldsymbol{\mathcal{A}}\|_1 = \sum_{i,j,k} |\boldsymbol{\mathcal{A}}_{i,j,k}|$ and $\|\boldsymbol{\mathcal{A}}\|_F = \sqrt{\sum_{i,j,k} |\boldsymbol{\mathcal{A}}_{i,j,k}|^2}$, respectively. For a matrix $\mathbf{A}$, its $(i, j)$-th entry is denoted as $\mathbf{A}_{i,j}$. The $i$-th row and $i$-th column of $\mathbf{A}$ are denoted by $\mathbf{a}_{(i)}$ and $\mathbf{a}_i$, respectively. The conjugate transpose of a matrix $\mathbf{A}$ is denoted by $\mathbf{A}^H$. The $\ell_{2,\infty}$ norm of $\mathbf{A}$ is defined as the maximum $\ell_2$ row norm, i.e., $\|\mathbf{A}\|_{2,\infty} = \max_i \|\mathbf{a}_{(i)}\|_2$. The matrix nuclear norm of $\mathbf{A}$ is $\|\mathbf{A}\|_* = \sum_i \sigma_i(\mathbf{A})$, where $\sigma_i(\mathbf{A})$'s are the singular values of $\mathbf{A}$.

Besides, for a tensor $\boldsymbol{\mathcal{A}} \in \mathbb{R}^{n_1 \times n_2 \times n_3}$, we define the block vectorizing and its inverse operation as $\texttt{bvec}(\boldsymbol{\mathcal{A}}) = [\mathbf{A}^{(1)}; \mathbf{A}^{(2)}; \cdots ; \mathbf{A}^{(n_3)}] \in \mathbb{R}^{n_1 n_3 \times n_2}$ and $\texttt{bvfold}(\texttt{bvec}(\boldsymbol{\mathcal{A}})) = \boldsymbol{\mathcal{A}}$, respectively. We denote $\bar{\mathbf{A}} \in \mathbb{C}^{n_1 n_3 \times n_2 n_3}$ as a block diagonal matrix with its $i$-th block on diagonal being the $i$-th frontal slice of $\bar{\boldsymbol{\mathcal{A}}}$, i.e.,

$$\bar{\mathbf{A}} = \texttt{bdiag}(\bar{\boldsymbol{\mathcal{A}}}) = \begin{bmatrix} \bar{\mathbf{A}}^{(1)} & & \\ & \ddots & \\ & & \bar{\mathbf{A}}^{(n_3)} \end{bmatrix}.$$

Finally, the block circulant matrix $\texttt{bcirc}(\boldsymbol{\mathcal{A}}) \in \mathbb{R}^{n_1 n_3 \times n_2 n_3}$ is defined as

$$\texttt{bcirc}(\boldsymbol{\mathcal{A}}) = \begin{bmatrix} \mathbf{A}^{(1)} & \mathbf{A}^{(n_3)} & \ldots & \mathbf{A}^{(2)} \\ \mathbf{A}^{(2)} & \mathbf{A}^{(1)} & \ldots & \mathbf{A}^{(3)} \\ \vdots & \vdots & \ddots & \vdots \\ \mathbf{A}^{(n_3)} & \mathbf{A}^{(n_3-1)} & \ldots & \mathbf{A}^{(1)} \end{bmatrix}.$$

**Definition 1** (t-product (Kilmer & Martin, 2011)). *The t-product between two tensors $\boldsymbol{\mathcal{A}} \in \mathbb{R}^{n_1 \times n_2 \times n_3}$ and $\boldsymbol{\mathcal{B}} \in \mathbb{R}^{n_2 \times n_4 \times n_3}$ is defined as*

$$\boldsymbol{\mathcal{C}} = \boldsymbol{\mathcal{A}} * \boldsymbol{\mathcal{B}} = \texttt{bvfold}(\texttt{bcirc}(\boldsymbol{\mathcal{A}}) \cdot \texttt{bvec}(\boldsymbol{\mathcal{B}})) \in \mathbb{R}^{n_1 \times n_4 \times n_3}. \tag{1}$$

The t-product in the spatial domain corresponds to matrix multiplication of the frontal slices in the Fourier domain; that is, $\boldsymbol{\mathcal{C}} = \boldsymbol{\mathcal{A}} * \boldsymbol{\mathcal{B}}$ is equivalent to $\bar{\mathbf{C}} = \bar{\mathbf{A}}\bar{\mathbf{B}}$ (Kilmer & Martin, 2011). Note that when $n_3 = 1$, the operator $*$ reduces to matrix multiplication.

---

**Algorithm 1** t-SVD for third-order tensors

---

**Input:** $\mathcal{A} \in \mathbb{R}^{n_1 \times n_2 \times n_3}$.

1: $\bar{\mathcal{A}} = \text{fft}(\mathcal{A}, [\,], 3)$.
2: **for** $k = 1, \ldots, n_3$ **do**
3:      $[\mathbf{U}, \mathbf{S}, \mathbf{V}] = \text{SVD}(\bar{\mathbf{A}}^{(k)})$.
4:      $\bar{\mathbf{U}}^{(k)} = \mathbf{U}$, $\bar{\mathbf{S}}^{(k)} = \mathbf{S}$, $\bar{\mathbf{V}}^{(k)} = \mathbf{V}$.
5: **end for**
6: $\mathcal{U} = \text{ifft}(\bar{\mathcal{U}}, [\,], 3)$, $\mathcal{S} = \text{ifft}(\bar{\mathcal{S}}, [\,], 3)$, $\mathcal{V} = \text{ifft}(\bar{\mathcal{V}}, [\,], 3)$.

**Output:** $\mathcal{U} \in \mathbb{R}^{n_1 \times n_1 \times n_3}$, $\mathcal{V} \in \mathbb{R}^{n_2 \times n_2 \times n_3}$ and $\mathcal{S} \in \mathbb{R}^{n_1 \times n_2 \times n_3}$ such that $\mathcal{A} = \mathcal{U} * \mathcal{S} * \mathcal{V}^T$.

---

**Definition 2** (Identity tensor (Kilmer & Martin, 2011)). *The identity tensor $\mathcal{I}_n \in \mathbb{R}^{n \times n \times n_3}$ is a tensor whose first frontal slice is the $n \times n$ identity matrix and all other frontal slices are zeros.*

**Definition 3** (Orthogonal tensor (Kilmer & Martin, 2011)). *A tensor $\mathcal{Q} \in \mathbb{R}^{n \times n \times n_3}$ is orthogonal if $\mathcal{Q} * \mathcal{Q}^T = \mathcal{Q}^T * \mathcal{Q} = \mathcal{I}_n$.*

Now we introduce the t-SVD for third-order tensors.

**Definition 4** (t-SVD (Kilmer et al., 2013)). *Let $\mathcal{A} \in \mathbb{R}^{n_1 \times n_2 \times n_3}$, then it can be factorized as $\mathcal{A} = \mathcal{U} * \mathcal{S} * \mathcal{V}^T$, where $\mathcal{U} \in \mathbb{R}^{n_1 \times n_1 \times n_3}$ and $\mathcal{V} \in \mathbb{R}^{n_2 \times n_2 \times n_3}$ are orthogonal tensors and $\mathcal{S} \in \mathbb{R}^{n_1 \times n_2 \times n_3}$ is a tensor whose frontal slices are diagonal matrices.*

One can obtain t-SVD efficiently by performing matrix SVDs in the Fourier domain as shown in Algorithm 1. Again, t-SVD reduces to the matrix SVD when $n_3 = 1$.

**Definition 5** (Tensor average and tubal rank (Lu et al., 2020)). *For any $\mathcal{A} \in \mathbb{R}^{n_1 \times n_2 \times n_3}$, the tensor average rank of $\mathcal{A}$ is defined as*

$$\text{rank}_a(\mathcal{A}) \stackrel{\text{def}}{=} \frac{1}{n_3}\text{rank}(\bar{\mathbf{A}}) = \frac{1}{n_3}\text{rank}(\text{bcirc}(\mathcal{A})).$$

*The tensor tubal rank $\text{rank}_t(\mathcal{A})$ is defined as the number of nonzero singular tubes of $\mathcal{S}$, i.e.,*

$$\text{rank}_t(\mathcal{A}) = \#\{i : \mathcal{S}(i, i, :) \neq \mathbf{0}\},$$

*where $\mathcal{S}$ is from the t-SVD of $\mathcal{A} = \mathcal{U} * \mathcal{S} * \mathcal{V}^T$.*

**Definition 6** (Tensor nuclear norm (Lu et al., 2020)). *Let $\mathcal{A} = \mathcal{U} * \mathcal{S} * \mathcal{V}^T$ be the t-SVD of $\mathcal{A} \in \mathbb{R}^{n_1 \times n_2 \times n_3}$. The tensor nuclear norm of $\mathcal{A}$ is defined as $\|\mathcal{A}\|_* = \langle \mathcal{S}, \mathcal{I} \rangle = \sum_{i=1}^r \mathcal{S}_{i,i,1}$, where $r = \text{rank}_t(\mathcal{A})$.*

It is known that the tensor nuclear norm is the convex envelope of the tensor average rank within the unit ball of the tensor spectral norm (Lu et al., 2020).

## 4 Method

In this section, we describe our approach for online max-norm regularized tensor decomposition. We begin our discussion by introducing our proposed tensor max-norm and mathematically formulating the problem.

### 4.1 Problem Formulation

In this paper, we consider the problem of online recovering of low tubal rank tensor from sparsely corrupted observations. Before going to the online scenario, we start with the batch setting. Suppose we are given a third-order tensor $\mathcal{Z} \in \mathbb{R}^{n_1 \times N \times n_3}$ consisting of $N$ samples that can be decomposed as a low tubal-rank component $\mathcal{X}$ and a sparse noise tensor $\mathcal{E}$. Our goal is to estimate the two components $\mathcal{X}$ and $\mathcal{E}$ by solving the following convex program:

$$\min_{\mathcal{X}, \mathcal{E}} \|\mathcal{X}\|_{\max}^2 + \lambda_1 \|\mathcal{E}\|_1 \quad \text{s.t.} \quad \mathcal{Z} = \mathcal{X} + \mathcal{E}, \tag{2}$$

where $\|\cdot\|_{\max}$ is the tensor max-norm defined later that encourages $\boldsymbol{\mathcal{X}}$ to be low-rank, and $\lambda_1 > 0$ is a penalty parameter.

Recall that both the matrix max and nuclear norms are alternative convex surrogates for the rank of a matrix. Bounding the matrix rank corresponds to constraining the dimensionality of each row of $\mathbf{L}$ and $\mathbf{R}$ in the factorization $\mathbf{X} = \mathbf{L}\mathbf{R}^H$ (Srebro & Shraibman, 2005). The nuclear norm encourages low-rank approximation by constraining the average row-norms of $\mathbf{L}$ and $\mathbf{R}$, whereas the max-norm, which is defined as $\|\mathbf{X}\|_{\max} = \min_{\mathbf{L},\mathbf{R}:\mathbf{X}=\mathbf{L}\mathbf{R}^H} \|\mathbf{L}\|_{2,\infty}\|\mathbf{R}\|_{2,\infty}$, promotes low-rankness by constraining the maximal row-norms of $\mathbf{L}$ and $\mathbf{R}$. Note that the t-product in the spatial domain corresponds to matrix multiplication in the Fourier domain, i.e., $\bar{\mathbf{X}} = \bar{\mathbf{L}}\bar{\mathbf{R}}^H$. Based on the property that $\|\boldsymbol{\mathcal{X}}\|_* = \frac{1}{n_3}\|\bar{\mathbf{X}}\|_*$ (Lu et al., 2020) and the matrix nuclear norm has an equivalent form $\|\mathbf{X}\|_* = \min_{\mathbf{L},\mathbf{R}:\mathbf{X}=\mathbf{L}\mathbf{R}^H} \frac{1}{2}(\|\mathbf{L}\|_F^2 + \|\mathbf{R}\|_F^2)$ (Recht et al., 2010), the tensor nuclear norm can be written as the following tensor factorization form:

$$\|\boldsymbol{\mathcal{X}}\|_* = \min_{\substack{\boldsymbol{\mathcal{L}}\in\mathbb{R}^{n_1\times r\times n_3} \\ \boldsymbol{\mathcal{R}}\in\mathbb{R}^{N\times r\times n_3}}} \{\frac{1}{2}(\|\boldsymbol{\mathcal{L}}\|_F^2 + \|\boldsymbol{\mathcal{R}}\|_F^2) = \frac{1}{2n_3}(\|\bar{\mathbf{L}}\|_F^2 + \|\bar{\mathbf{R}}\|_F^2) : \boldsymbol{\mathcal{X}} = \boldsymbol{\mathcal{L}} * \boldsymbol{\mathcal{R}}^T\}, \tag{3}$$

where $r$ is an upper bound on the tensor tubal rank of $\boldsymbol{\mathcal{X}}$. As a theoretical foundation of the work Srebro et al. (2004), Srebro & Shraibman (2005) provided the relationships between the rank, nuclear norm and max-norm as complexity measures of matrices, which gave us some intuition on why the max-norm regularizer could outperform the nuclear-norm regularizer in some applications, e.g., Lee et al. (2010) and Fang et al. (2018). Motivated by this advantage, we propose to define the max-norm of the tensor $\boldsymbol{\mathcal{X}} \in \mathbb{R}^{n_1\times N\times n_3}$ using the same tensor factorization form by constraining all rows of $\bar{\mathbf{L}}$ and $\bar{\mathbf{R}}$ to have small $\ell_2$ norms as follows:

**Definition 7** (Tensor max-norm). *The tensor max-norm of $\boldsymbol{\mathcal{X}} \in \mathbb{R}^{n_1\times N\times n_3}$ is defined as*

$$\|\boldsymbol{\mathcal{X}}\|_{\max} \overset{\text{def}}{=} \min_{\substack{\boldsymbol{\mathcal{L}}\in\mathbb{R}^{n_1\times r\times n_3} \\ \boldsymbol{\mathcal{R}}\in\mathbb{R}^{N\times r\times n_3}}} \{\|\bar{\mathbf{L}}\|_{2,\infty}\cdot\|\bar{\mathbf{R}}\|_{2,\infty} : \boldsymbol{\mathcal{X}} = \boldsymbol{\mathcal{L}} * \boldsymbol{\mathcal{R}}^T\}, \tag{4}$$

*where $r$ is an upper bound on the tensor tubal rank of $\boldsymbol{\mathcal{X}}$. We also have $\|\boldsymbol{\mathcal{X}}\|_{\max} = \|\bar{\mathbf{X}}\|_{\max}$.*

Now assume that the samples $\boldsymbol{\mathcal{Z}}(:,i,:)$, $i = 1,\dots,N$, are observed sequentially, our objective is to efficiently learn the low-rank component $\boldsymbol{\mathcal{X}}$ and error tensor $\boldsymbol{\mathcal{E}}$ in an online fashion. To facilitate online optimization, instead of solving the constrained problem (2) directly, we relax the constraint by regarding it as a quadratic penalty, resulting in

$$\min_{\boldsymbol{\mathcal{L}},\boldsymbol{\mathcal{R}},\boldsymbol{\mathcal{E}}} \frac{1}{2}\|\boldsymbol{\mathcal{Z}} - \boldsymbol{\mathcal{L}} * \boldsymbol{\mathcal{R}}^T - \boldsymbol{\mathcal{E}}\|_F^2 + \frac{\lambda_1}{2}\|\bar{\mathbf{L}}\|_{2,\infty}^2\|\bar{\mathbf{R}}\|_{2,\infty}^2 + \lambda_2\|\boldsymbol{\mathcal{E}}\|_1, \tag{5}$$

where $\lambda_1$ and $\lambda_2$ balance the importance of each term in (5). Intuitively, the variable $\boldsymbol{\mathcal{L}}$ corresponds to a basis for the clean data and each horizontal slice of $\boldsymbol{\mathcal{R}}$ corresponds to the coefficients associated with each sample. Notice that the size of the coefficient tensor $\boldsymbol{\mathcal{R}}$ is proportional to $N$. In order to compute the optimal coefficients for the $i$-th sample, we need to compute the gradient of $\|\bar{\mathbf{R}}\|_{2,\infty}$. Moreover, each horizontal slice of $\boldsymbol{\mathcal{R}}$ corresponds to one sample, hence the computation of such gradient requires to access all the data. Fortunately, we have the following proposition that alleviates the inter-dependencies among the horizontal slices of $\boldsymbol{\mathcal{R}}$, so that we can optimize them sequentially.

**Proposition 1.** *Problem* (5) *is equivalent to the following constrained program:*

$$\min_{\boldsymbol{\mathcal{L}},\boldsymbol{\mathcal{R}},\boldsymbol{\mathcal{E}}} \frac{1}{2}\|\boldsymbol{\mathcal{Z}} - \boldsymbol{\mathcal{L}} * \boldsymbol{\mathcal{R}}^T - \boldsymbol{\mathcal{E}}\|_F^2 + \frac{\lambda_1}{2}\|\bar{\mathbf{L}}\|_{2,\infty}^2 + \lambda_2\|\boldsymbol{\mathcal{E}}\|_1 \quad \text{s.t.} \quad \|\bar{\mathbf{R}}\|_{2,\infty}^2 \le 1. \tag{6}$$

*Here, "equivalent" means the optimal values of the objective functions in* (5) *and* (6) *are the same. Moreover, there exists an optimal solution $(\boldsymbol{\mathcal{L}}^\star, \boldsymbol{\mathcal{R}}^\star, \boldsymbol{\mathcal{E}}^\star)$ attained such that $\|\bar{\mathbf{R}}^\star\|_{2,\infty} = 1$.*

In the constrained program (6), the coefficients associated with each individual sample (i.e., one horizontal slice of the coefficient tensor) are now uniformly and separately constrained. Let $\overrightarrow{\boldsymbol{\mathcal{Z}}}_i$, $\overrightarrow{\boldsymbol{\mathcal{R}}}_i$ and $\overrightarrow{\boldsymbol{\mathcal{E}}}_i$ be the $i$-th lateral slices of tensors $\boldsymbol{\mathcal{Z}}$, $\boldsymbol{\mathcal{R}}^T$ and $\boldsymbol{\mathcal{E}}$, respectively. We define

$$\widetilde{\ell}(\overrightarrow{\boldsymbol{\mathcal{Z}}}, \boldsymbol{\mathcal{L}}, \overrightarrow{\boldsymbol{\mathcal{R}}}, \overrightarrow{\boldsymbol{\mathcal{E}}}) \overset{\text{def}}{=} \frac{1}{2}\|\overrightarrow{\boldsymbol{\mathcal{Z}}} - \boldsymbol{\mathcal{L}} * \overrightarrow{\boldsymbol{\mathcal{R}}} - \overrightarrow{\boldsymbol{\mathcal{E}}}\|_F^2 + \lambda_2\|\overrightarrow{\boldsymbol{\mathcal{E}}}\|_1.$$

---

**Algorithm 2** Online Max-Norm Regularized Tensor Decomposition

---

**Input:** Observed samples $\boldsymbol{\mathcal{Z}} \in \mathbb{R}^{n_1 \times N \times n_3}$, and parameters $\lambda_1$ and $\lambda_2$.
**Initialize:** Random basis $\boldsymbol{\mathcal{L}}_0 \in \mathbb{R}^{n_1 \times r \times n_3}$, $\boldsymbol{\mathcal{A}}_0 = \boldsymbol{\mathcal{B}}_0 = \mathbf{0}$.

1: **for** $t = 1, 2, \ldots, N$ **do**
2:     Access the $t$-th sample $\vec{\boldsymbol{\mathcal{Z}}}_t$.
3:     Update $\{\vec{\boldsymbol{\mathcal{R}}}_t^\star, \vec{\boldsymbol{\mathcal{E}}}_t^\star\}$ by solving (10).
4:     Update the accumulation tensors:

$$\boldsymbol{\mathcal{A}}_t = \boldsymbol{\mathcal{A}}_{t-1} + \vec{\boldsymbol{\mathcal{R}}}_t^\star * \vec{\boldsymbol{\mathcal{R}}}_t^{\star T},$$
$$\boldsymbol{\mathcal{B}}_t = \boldsymbol{\mathcal{B}}_{t-1} + (\vec{\boldsymbol{\mathcal{Z}}}_t - \vec{\boldsymbol{\mathcal{E}}}_t^\star) * \vec{\boldsymbol{\mathcal{R}}}_t^{\star T}.$$

5:     Update the basis $\boldsymbol{\mathcal{L}}_t$ (equivalently, $\overline{\mathbf{L}}_t$) by optimizing the surrogate function:

$$\overline{\mathbf{L}}_t = \arg\min_{\overline{\mathbf{L}}} \frac{1}{tn_3} \left( \frac{1}{2} \mathrm{tr}\big(\overline{\mathbf{L}}^H \overline{\mathbf{L}} \overline{\mathbf{A}}_t\big) - \mathrm{tr}\big(\overline{\mathbf{L}}^H \overline{\mathbf{B}}_t\big) \right) + \frac{\lambda_1}{2t} \|\overline{\mathbf{L}}\|_{2,\infty}^2.$$

6: **end for**
**Output:** Optimal basis $\boldsymbol{\mathcal{L}}_N$.

---

Equipped with Proposition 1, we can rewrite the original problem (5) as the following, with each sample being separately processed:

$$\min_{\boldsymbol{\mathcal{L}}} \min_{\boldsymbol{\mathcal{R}}, \boldsymbol{\mathcal{E}}} \sum_{i=1}^{N} \widetilde{\ell}(\vec{\boldsymbol{\mathcal{Z}}}_i, \boldsymbol{\mathcal{L}}, \vec{\boldsymbol{\mathcal{R}}}_i, \vec{\boldsymbol{\mathcal{E}}}_i) + \frac{\lambda_1}{2} \|\overline{\mathbf{L}}\|_{2,\infty}^2 \quad \text{s.t.} \quad \forall i = 1, \ldots, N, \quad k = 1, \ldots, n_3, \quad \|\vec{\overline{\mathbf{R}}}_i^{(k)}\|_2^2 \leq 1, \tag{7}$$

where $\vec{\overline{\boldsymbol{\mathcal{R}}}}_i = \texttt{fft}(\vec{\boldsymbol{\mathcal{R}}}_i, [\,], 3)$ and $\vec{\overline{\mathbf{R}}}_i^{(k)}$ denotes the $k$-th frontal slice of $\vec{\overline{\boldsymbol{\mathcal{R}}}}_i$. This is indeed equivalent to optimizing the *empirical loss* function:

$$f_N(\boldsymbol{\mathcal{L}}) \stackrel{\text{def}}{=} \frac{1}{N} \sum_{i=1}^{N} \ell(\boldsymbol{\mathcal{L}}, \vec{\boldsymbol{\mathcal{Z}}}_i) + \frac{\lambda_1}{2N} \|\overline{\mathbf{L}}\|_{2,\infty}^2, \tag{8}$$

where $\ell(\boldsymbol{\mathcal{L}}, \vec{\boldsymbol{\mathcal{Z}}}) = \min_{\vec{\boldsymbol{\mathcal{R}}} \in \mathscr{P}, \vec{\boldsymbol{\mathcal{E}}}} \widetilde{\ell}(\vec{\boldsymbol{\mathcal{Z}}}, \boldsymbol{\mathcal{L}}, \vec{\boldsymbol{\mathcal{R}}}, \vec{\boldsymbol{\mathcal{E}}})$ with $\mathscr{P} = \{\vec{\boldsymbol{\mathcal{R}}} \in \mathbb{R}^{r \times 1 \times n_3} : \forall k, \|\vec{\overline{\mathbf{R}}}^{(k)}\|_2^2 \leq 1\}$. We assume that each sample is drawn independently and identically distributed from some unknown distribution. In stochastic optimization, one is usually interested in the minimization of the *expected loss* function

$$f(\boldsymbol{\mathcal{L}}) \stackrel{\text{def}}{=} \mathbb{E}[\ell(\boldsymbol{\mathcal{L}}, \vec{\boldsymbol{\mathcal{Z}}})] = \lim_{N \to +\infty} f_N(\boldsymbol{\mathcal{L}}). \tag{9}$$

In this work, we first establish a surrogate function for this expected cost and then optimize the surrogate function for obtaining the basis in an online fashion.

## 4.2 Online Implementation of OMRTD

We now describe our online algorithm for minimizing the empirical loss function (8). The detailed algorithm is summarized in Algorithm 2. We start with an initial random dictionary $\boldsymbol{\mathcal{L}}_0$. The key idea is that at each iteration $t$, we first minimize the loss function with respect to $\{\vec{\boldsymbol{\mathcal{R}}}_t, \vec{\boldsymbol{\mathcal{E}}}_t\}$ given the previous $\boldsymbol{\mathcal{L}}_{t-1}$ by solving (7), and then further refine the dictionary $\boldsymbol{\mathcal{L}}_t$ by minimizing the cumulative loss.

**Optimize $\vec{\boldsymbol{\mathcal{R}}}$ and $\vec{\boldsymbol{\mathcal{E}}}$:** Given $\boldsymbol{\mathcal{L}}_{t-1}$ in the previous iteration, we obtain the optimal solution $\{\vec{\boldsymbol{\mathcal{R}}}_t^\star, \vec{\boldsymbol{\mathcal{E}}}_t^\star\}$ by solving

$$\{\vec{\boldsymbol{\mathcal{R}}}_t^\star, \vec{\boldsymbol{\mathcal{E}}}_t^\star\} = \arg\min_{\vec{\boldsymbol{\mathcal{R}}} \in \mathscr{P}, \vec{\boldsymbol{\mathcal{E}}}} \frac{1}{2} \|\vec{\boldsymbol{\mathcal{Z}}}_t - \boldsymbol{\mathcal{L}}_{t-1} * \vec{\boldsymbol{\mathcal{R}}} - \vec{\boldsymbol{\mathcal{E}}}\|_F^2 + \lambda_2 \|\vec{\boldsymbol{\mathcal{E}}}\|_1. \tag{10}$$

We employ a block coordinate descent algorithm which alternatively updates one variable at a time with the other being fixed until some stopping criteria is satisfied. For the sake of exposition, we omit the subscript $t$ here. According to Bertsekas (1999), this alternating minimization procedure is guaranteed to converge when the objective function is strongly convex with respect to each block variable. It can be observed from (10) that the strong convexity holds for $\overrightarrow{\boldsymbol{\mathcal{E}}}$ and it holds for $\overrightarrow{\boldsymbol{\mathcal{R}}}$ if and only if $\boldsymbol{\mathcal{L}}$ is full tubal rank. For computational efficiency, we actually append a small jitter $\frac{\epsilon}{2}\|\overrightarrow{\boldsymbol{\mathcal{R}}}\|_F^2$ to the objective if necessary so as to guarantee the convergence (we set $\epsilon = 0.01$ in our experiments). We first compute an initial guess of $\overrightarrow{\boldsymbol{\mathcal{R}}}$ by

$$\overrightarrow{\boldsymbol{\mathcal{R}}}_{\text{cand}} = (\boldsymbol{\mathcal{L}}^T * \boldsymbol{\mathcal{L}} + \epsilon \boldsymbol{\mathcal{I}}_r)^{-1} * \boldsymbol{\mathcal{L}}^T * (\overrightarrow{\boldsymbol{\mathcal{Z}}} - \overrightarrow{\boldsymbol{\mathcal{E}}}). \tag{11}$$

If this initial guess $\overrightarrow{\boldsymbol{\mathcal{R}}}_{\text{cand}}$ satisfies the constraint that $\overrightarrow{\boldsymbol{\mathcal{R}}}_{\text{cand}} \in \mathscr{P}$, we will use $\overrightarrow{\boldsymbol{\mathcal{R}}}_{\text{cand}}$ as the new iterate for $\overrightarrow{\boldsymbol{\mathcal{R}}}$. Otherwise, for each $k$, if $\|\overrightarrow{\mathbf{R}}_{\text{cand}}^{(k)}\|_2 > 1$, we will introduce a positive dual variable $\eta^{(k)}$ and solve

$$\max_{\eta^{(k)}} \min_{\overrightarrow{\mathbf{R}}^{(k)}} \frac{1}{2}\|\overrightarrow{\mathbf{Z}}^{(k)} - \overline{\mathbf{L}}\overrightarrow{\mathbf{R}}^{(k)} - \overrightarrow{\mathbf{E}}^{(k)}\|_2^2 + \frac{\eta^{(k)}}{2}(\|\overrightarrow{\mathbf{R}}^{(k)}\|_2^2 - 1) \quad \text{s.t.} \quad \eta^{(k)} > 0, \|\overrightarrow{\mathbf{R}}^{(k)}\|_2 = 1. \tag{12}$$

The closed-form solution of (12) is given by

$$\overrightarrow{\mathbf{R}}^{(k)} = (\overline{\mathbf{L}}^{(k)^H}\overline{\mathbf{L}}^{(k)} + \eta^{(k)}\mathbf{I}_r)^{-1}\overline{\mathbf{L}}^{(k)^H}(\overrightarrow{\mathbf{Z}}^{(k)} - \overrightarrow{\mathbf{E}}^{(k)}),$$

where $\mathbf{I}_r$ denotes the $r \times r$ identity matrix. According to Proposition 2 of Shen et al. (2017), for each $k$, $\|\overrightarrow{\mathbf{R}}^{(k)}\|_2$ is a strictly monotonically decreasing function with respect to $\eta^{(k)}$. This allows us to search the optimal $\overrightarrow{\mathbf{R}}^{(k)}$ as well as the dual variable $\eta^{(k)}$ using bisection method. To be concrete, we denote

$$\overrightarrow{\mathbf{R}}^{(k)}(\eta) = (\overline{\mathbf{L}}^{(k)^H}\overline{\mathbf{L}}^{(k)} + \eta\mathbf{I}_r)^{-1}\overline{\mathbf{L}}^{(k)^H}(\overrightarrow{\mathbf{Z}}^{(k)} - \overrightarrow{\mathbf{E}}^{(k)})$$

and maintain a lower bound $\eta_1$ and an upper bound $\eta_2$ such that $\|\overrightarrow{\mathbf{R}}^{(k)}(\eta_1)\|_2 \geq 1$ and $\|\overrightarrow{\mathbf{R}}^{(k)}(\eta_2)\|_2 \leq 1$. This ensures that the optimal $\eta^{(k)}$ lies within the interval $[\eta_1, \eta_2]$ and we can find this value efficiently using bisection method outlined in Algorithm 5 in Appendix A.2. By comparing $\|\overrightarrow{\mathbf{R}}^{(k)}(\eta^{(k)})\|_2$ at $\eta^{(k)} = (\eta_1 + \eta_2)/2$ with 1, we can either increase $\eta_1$ or decrease $\eta_2$ until $\|\overrightarrow{\mathbf{R}}^{(k)}(\eta^{(k)})\|_2$ equals one. Note that $\|\overrightarrow{\mathbf{R}}_{\text{cand}}^{(k)}\|_2 > 1$, which implies that the optimal value for $\eta^{(k)}$ is greater than $\epsilon$, thus we can easily set $\eta_1 = 0$. Once $\overrightarrow{\boldsymbol{\mathcal{R}}}$ is found, we can update $\overrightarrow{\boldsymbol{\mathcal{E}}}$ using the soft-thresholding operator (Donoho, 1995): $\overrightarrow{\boldsymbol{\mathcal{E}}} = S_{\lambda_2}[\overrightarrow{\boldsymbol{\mathcal{Z}}} - \boldsymbol{\mathcal{L}} * \overrightarrow{\boldsymbol{\mathcal{R}}}]$. For completeness, we list our method for updating the coefficient tensor and the noise tensor in Algorithm 4 in Appendix A.2.

**Optimize $\boldsymbol{\mathcal{L}}$:** When $\{\overrightarrow{\boldsymbol{\mathcal{Z}}}_i, \overrightarrow{\boldsymbol{\mathcal{R}}}_i^\star, \overrightarrow{\boldsymbol{\mathcal{E}}}_i^\star\}_{i=1}^t$ are available, we can update the basis $\boldsymbol{\mathcal{L}}_t$ by optimizing the following objective function

$$g_t(\boldsymbol{\mathcal{L}}) = \frac{1}{t}\sum_{i=1}^t \widetilde{\ell}(\overrightarrow{\boldsymbol{\mathcal{Z}}}_i, \boldsymbol{\mathcal{L}}, \overrightarrow{\boldsymbol{\mathcal{R}}}_i^\star, \overrightarrow{\boldsymbol{\mathcal{E}}}_i^\star) + \frac{\lambda_1}{2t}\|\overline{\mathbf{L}}\|_{2,\infty}^2. \tag{13}$$

This is a surrogate function of the empirical cost function $f_t(\boldsymbol{\mathcal{L}})$ defined in (9) in a sense that it provides an upper bound for $f_t(\boldsymbol{\mathcal{L}})$: $g_t(\boldsymbol{\mathcal{L}}) \geq f_t(\boldsymbol{\mathcal{L}})$ (Mairal et al., 2010; Feng et al., 2013; Shen et al., 2017). It is easy to verify that the minimizer of (13) is given by

$$\boldsymbol{\mathcal{L}}_t = \arg\min_{\boldsymbol{\mathcal{L}}} \frac{1}{2t}\|\boldsymbol{\mathcal{Z}}_t - \boldsymbol{\mathcal{L}} * \boldsymbol{\mathcal{R}}_t^\star - \boldsymbol{\mathcal{E}}_t^\star\|_F^2 + \frac{\lambda_1}{2t}\|\overline{\mathbf{L}}\|_{2,\infty}^2,$$

where $\boldsymbol{\mathcal{Z}}_t = [\overrightarrow{\boldsymbol{\mathcal{Z}}}_1, \ldots, \overrightarrow{\boldsymbol{\mathcal{Z}}}_t] \in \mathbb{R}^{n_1 \times t \times n_3}$, $\boldsymbol{\mathcal{R}}_t^\star = [\overrightarrow{\boldsymbol{\mathcal{R}}}_1^\star, \ldots, \overrightarrow{\boldsymbol{\mathcal{R}}}_t^\star] \in \mathbb{R}^{r \times t \times n_3}$ and $\boldsymbol{\mathcal{E}}_t^\star = [\overrightarrow{\boldsymbol{\mathcal{E}}}_1^\star, \ldots, \overrightarrow{\boldsymbol{\mathcal{E}}}_t^\star] \in \mathbb{R}^{n_1 \times t \times n_3}$. Let $\overline{\mathbf{L}}_t = \texttt{bdiag}(\overline{\boldsymbol{\mathcal{L}}}_t)$, $\overline{\mathbf{Z}}_t = \texttt{bdiag}(\overline{\boldsymbol{\mathcal{Z}}}_t)$, $\overline{\mathbf{R}}_t^\star = \texttt{bdiag}(\overline{\boldsymbol{\mathcal{R}}}_t^\star)$ and $\overline{\mathbf{E}}_t^\star = \texttt{bdiag}(\overline{\boldsymbol{\mathcal{E}}}_t^\star)$. The above problem can be

transformed into the Fourier domain as

$$\bar{\mathbf{L}}_t = \arg\min_{\bar{\mathbf{L}}} \frac{1}{2tn_3} \|\bar{\mathbf{Z}}_t - \bar{\mathbf{L}}\bar{\mathbf{R}}_t^\star - \bar{\mathbf{E}}_t^\star\|_F^2 + \frac{\lambda_1}{2t}\|\bar{\mathbf{L}}\|_{2,\infty}^2$$

$$= \arg\min_{\bar{\mathbf{L}}} \frac{1}{tn_3}\left(\frac{1}{2}\mathrm{tr}(\bar{\mathbf{L}}^H\bar{\mathbf{L}}\bar{\mathbf{A}}_t) - \mathrm{tr}(\bar{\mathbf{L}}^H\bar{\mathbf{B}}_t)\right) + \frac{\lambda_1}{2t}\|\bar{\mathbf{L}}\|_{2,\infty}^2. \tag{14}$$

Here, $\boldsymbol{\mathcal{A}}_t = \sum_{i=1}^t \overrightarrow{\boldsymbol{\mathcal{R}}}_i^\star * \overrightarrow{\boldsymbol{\mathcal{R}}}_i^{\star T} \in \mathbb{R}^{r \times r \times n_3}$, $\boldsymbol{\mathcal{B}}_t = \sum_{i=1}^t (\overrightarrow{\boldsymbol{\mathcal{Z}}}_i - \overrightarrow{\boldsymbol{\mathcal{E}}}_i^\star) * \overrightarrow{\boldsymbol{\mathcal{R}}}_i^{\star T} \in \mathbb{R}^{n_1 \times r \times n_3}$ and $\mathrm{tr}(\cdot)$ denotes the trace operation. We again omit the subscript $t$ in the rest of this paragraph. In order to derive the optimal solution, we first need to compute the subgradient of the squared $\ell_{2,\infty}$ norm, which can be done in a similar way as in Shen et al. (2017). To be specific, let $\boldsymbol{\Theta}$ denote the set of row indices corresponding to the rows with maximum $\ell_2$ row norm of $\bar{\mathbf{L}}$. Define $\mathbf{Q} \in \mathbb{R}^{n_1 n_3 \times n_1 n_3}$ to be a positive semi-definite diagonal matrix with $\mathbf{Q}_{i,i} \neq 0$ if and only if $i \in \boldsymbol{\Theta}$ and all other entries are zeros such that $\sum_{i=1}^{n_1 n_3} \mathbf{Q}_{i,i} = 1$. The subgradient of $\frac{1}{2}\|\bar{\mathbf{L}}\|_{2,\infty}^2$ can be written as $\mathbf{H} = \partial(\frac{1}{2}\|\bar{\mathbf{L}}\|_{2,\infty}^2) = \mathbf{Q}\bar{\mathbf{L}}$. We then use block coordinate descent (Bertsekas, 1999) to update the columns of $\bar{\mathbf{L}}$ sequentially; see more details in Algorithm 6 in Appendix A.2. Note that $\mathbf{H}$ is a diagonal matrix and off-diagonal blocks are zero matrices, we can write $\mathbf{H}$ as

$$\mathbf{H} \overset{\mathrm{def}}{=} \begin{bmatrix} \mathbf{H}_1 & & \\ & \ddots & \\ & & \mathbf{H}_{n_3} \end{bmatrix}.$$

In this manner, the subgradient of the squared $\ell_{2,\infty}$ norm of $\bar{\mathbf{L}}$ with respect to $\bar{\mathbf{L}}^{(k)}$ is $\mathbf{H}_k$. By assuming that the objective function of (14) is strongly convex with respect to $\bar{\mathbf{L}}$, it is guaranteed that the solution of this block coordinate descent scheme always converges to the global optimum.

### 4.3 Complexity and Convergence Analysis

Here, we provide further analysis on complexity and convergence of our method.

**Computational Complexity.** In each iteration, the computational burden is dominated by the cost for solving (10). The computational complexity of (11) involves computing the inverse of $n_3$ $r \times r$ matrices, matrix multiplications, and the (inverse) Fast Fourier Transform, totally $\mathcal{O}(n_1 r^2 n_3 + n_1 n_3 \log n_3)$. For the basis update, obtaining a subgradient of the squared $\ell_{2,\infty}$ norm of $\bar{\mathbf{L}}$ is $\mathcal{O}(n_1 r n_3)$ and one-pass update for the columns in $\bar{\mathbf{L}}$ in Algorithm 6 requires $\mathcal{O}(n_1 r^2 n_3)$.

**Memory Cost.** OMRTD requires $\mathcal{O}(n_1 r n_3)$ to load $\boldsymbol{\mathcal{L}}_{t-1}$ and $\overrightarrow{\boldsymbol{\mathcal{Z}}}_t$ to obtain $\{\overrightarrow{\boldsymbol{\mathcal{R}}}_t, \overrightarrow{\boldsymbol{\mathcal{E}}}_t\}$. To store the accumulation tensor $\boldsymbol{\mathcal{A}}_t$, we need $\mathcal{O}(r^2 n_3)$ memory while that for $\boldsymbol{\mathcal{B}}_t$ is $\mathcal{O}(n_1 r n_3)$. Finally, we find that only $\boldsymbol{\mathcal{A}}_t$ and $\boldsymbol{\mathcal{B}}_t$ are needed for the computation of the new iterate $\boldsymbol{\mathcal{L}}_t$. Hence, the memory cost of OMRTD is $\mathcal{O}(n_1 r n_3)$, i.e., independent of $N$, making our algorithm appealing for large-scale streaming tensor data.

**Convergence.** In order to present the validity of the proposed algorithm, we make the following assumptions:

**(A1)** The observed samples are generated independent identically distributed from some distribution and there exist two positive constants $\alpha_0, \alpha_1$, such that the conditions $\alpha_0 \leq \|\overrightarrow{\boldsymbol{\mathcal{Z}}}_t\|_F \leq \alpha_1$ holds almost surely for all $t \geq 1$. This assumption is quite natural for the realistic data such as images and videos.

**(A2)** The surrogate function $g_t(\boldsymbol{\mathcal{L}})$ in (13) is strongly convex. Particularly, we assume that the smallest singular value of the matrix $\frac{1}{t}\bar{\mathbf{A}}_t$ is not smaller than some positive constant $\beta_1$.

**(A3)** The minimizer for $\ell(\boldsymbol{\mathcal{L}}, \overrightarrow{\boldsymbol{\mathcal{Z}}})$ is unique. Notice that $\widetilde{\ell}(\overrightarrow{\boldsymbol{\mathcal{Z}}}, \boldsymbol{\mathcal{L}}, \overrightarrow{\boldsymbol{\mathcal{R}}}, \overrightarrow{\boldsymbol{\mathcal{E}}})$ is strongly convex with respect to $\overrightarrow{\boldsymbol{\mathcal{E}}}$ and convex with respect to $\overrightarrow{\boldsymbol{\mathcal{R}}}$. We can enforce this assumption by adding a term $\frac{\epsilon}{2}\|\overrightarrow{\boldsymbol{\mathcal{R}}}\|_F^2$ to the objective function, where $\epsilon$ is a small positive constant.

Based on assumptions (A1), (A2) and (A3), we establish the main theoretical result of this work.

**Theorem 1.** *Assume (A1), (A2) and (A3). Let $\{\mathcal{L}_t\}_{t\geq1}$ be the solution produced by Algorithm 2. Then the sequence converges to a stationary point of the expected loss function $f(\mathcal{L})$ when $t$ tends to infinity.*

The proof of Theorem 1 proceeds in the same following four steps as in Shen et al. (2017):

(I) We first show that all the stochastic variables $\{\overrightarrow{\mathcal{R}}_t, \overrightarrow{\mathcal{E}}_t, \mathcal{L}_t\}_{t=1}^\infty$ are uniformly bounded. This property is important because it justifies that the problem we are solving is well-defined.

**Proposition 2.** *Let $\{\overrightarrow{\mathcal{R}}_t^\star, \overrightarrow{\mathcal{E}}_t^\star, \mathcal{L}_t\}_{t=1}^\infty$ be the sequence of optimal solutions produced by Algorithm 2. Then*

1. *The optimal solutions $\overrightarrow{\mathcal{R}}_t^\star$ and $\overrightarrow{\mathcal{E}}_t^\star$ are uniformly bounded.*

2. *The tensors $\frac{1}{t}\mathcal{A}_t$ and $\frac{1}{t}\mathcal{B}_t$ are uniformly bounded.*

3. *$\mathcal{L}_t$ is supported on some compact set $\mathscr{L}$, there exists a positive constant $c_1$, such that for all $t > 0$, we have $\|\mathcal{L}_t\|_F \leq c_1$.*

**Corollary 1.** *Let $\{\overrightarrow{\mathcal{R}}_t^\star, \overrightarrow{\mathcal{E}}_t^\star, \mathcal{L}_t\}_{t=1}^\infty$ be the sequence of optimal solutions produced by Algorithm 2. Then, for all $t \geq 1$, we have*

1. *$\widetilde{\ell}(\overrightarrow{\mathcal{Z}}_t, \mathcal{L}_t, \overrightarrow{\mathcal{R}}_t^\star, \overrightarrow{\mathcal{E}}_t^\star)$ and $\ell(\mathcal{L}_t, \overrightarrow{\mathcal{Z}}_t)$ are uniformly bounded from above.*

2. *The surrogate function, $g_t(\mathcal{L}_t)$ defined in (13), is uniformly bounded.*

3. *Moreover, $g_t(\mathcal{L}_t)$ is uniformly Lipschitz over the compact set $\mathscr{L}$.*

(II) Next, we show that the positive stochastic process $g_t(\mathcal{L}_t)$ defined in (13) converges almost surely. First, we can easily show that $g_{t+1}(\mathcal{L}_{t+1}) - g_t(\mathcal{L}_t)$ is upper bounded by $\frac{\ell(\mathcal{L}_t, \overrightarrow{\mathcal{Z}}_{t+1}) - f_t(\mathcal{L}_t)}{t+1}$. Then we show that the set of measurable functions $\{\ell(\mathcal{L}, \overrightarrow{\mathcal{Z}}), \mathcal{L} \in \mathscr{L}\}$ is P-Donsker (van der Vaart, 1998), and the difference between the empirical loss and expected loss can be uniformly upper bounded by $\mathcal{O}(1/\sqrt{t})$. Therefore, following the proof in Shen et al. (2017, Theorem 4), we conclude that $g_t(\mathcal{L}_t)$ is a quasi-martingale (Fisk, 1965) and converges almost surely.

(III) By establishing the numerical convergence of the basis sequence $\{\mathcal{L}_t\}_{t=1}^\infty$, i.e., $\|\mathcal{L}_{t+1} - \mathcal{L}_t\|_F = \mathcal{O}(1/t)$ (this step corresponds to Shen et al. (2017, Proposition 10)), we can show that the empirical loss function, $f_t(\mathcal{L}_t)$ defined in (8) converges almost surely to the same limit of its surrogate $g_t(\mathcal{L}_t)$. According to the central limit theorem, $f_t(\mathcal{L}_t)$ also converges almost surely to the expected loss $f(\mathcal{L}_t)$ defined in (9), implying that $g_t(\mathcal{L}_t)$ and $f(\mathcal{L}_t)$ converge to the same limit (this step corresponds to Shen et al. (2017, Theorem 5)).

(IV) Finally, using the expression $\mathcal{L} * \overrightarrow{\mathcal{R}} = \texttt{bvfold}(\texttt{bcirc}(\mathcal{L}) \cdot \texttt{bvec}(\overrightarrow{\mathcal{R}}))$, we can derive that $\nabla f(\mathcal{L})$ is uniformly Lipschitz on $\mathcal{L}$. Consequently, by taking a simple Taylor expansion, it justifies that the gradient of $f(\mathcal{L})$ taking at $\mathcal{L}_t$ vanishes as $t$ tends to infinity, which concludes Theorem 1.

### 4.4 Extension to Tensor Completion

In this subsection, we study OMRTD for the case of data having missing entries. To be specific, let $\mathcal{W} = [0, 1] \in \mathbb{R}^{n_1 \times N \times n_3}$ be a tensor such that $\mathcal{W}_{i,j,k} = 1$ if $\mathcal{Z}_{i,j,k}$ is observed and $\mathcal{W}_{i,j,k} = 0$ otherwise. The locations of the observed entries can be indexed by a set $\mathbf{\Omega} = \{(i, j, k) : \mathcal{W}_{i,j,k} = 1\}$. We reformulate the problem (2) for tensor completion by solving the following problem:

$$\min_{\mathcal{X}, \mathcal{E}} \frac{1}{2}\|\mathcal{M} - \mathcal{X} - \mathcal{E}\|_F^2 + \frac{\lambda_1}{2}\|\mathcal{X}\|_{\max}^2 + \lambda_2\|\mathcal{E}\|_1 \quad \text{s.t.} \quad P_{\mathbf{\Omega}}(\mathcal{M}) = P_{\mathbf{\Omega}}(\mathcal{Z}), \tag{15}$$

where $P_{\mathbf{\Omega}}(\cdot)$ is the orthogonal projector onto the span of tensors vanishing outside $\mathbf{\Omega}$ so that the $(i, j, k)$-th entry of $P_{\mathbf{\Omega}}(\mathcal{M})$ is equal to $\mathcal{M}_{i,j,k}$ if $(i, j, k) \in \mathbf{\Omega}$ and zero otherwise. We again minimize (15) by alternating

---

**Algorithm 3** Updating tensor columns of $\boldsymbol{\mathcal{M}}$, $\boldsymbol{\mathcal{R}}$ and $\boldsymbol{\mathcal{E}}$

---

**Input:** Partially observed data sample $P_{\boldsymbol{\Phi}}(\overrightarrow{\boldsymbol{\mathcal{Z}}}) \in \mathbb{R}^{n_1 \times 1 \times n_3}$, $\boldsymbol{\mathcal{L}} \in \mathbb{R}^{n_1 \times r \times n_3}$, and parameter $\lambda_2$.

**Initialize:** $\overrightarrow{\boldsymbol{\mathcal{R}}}^{(0)} = \overrightarrow{\boldsymbol{\mathcal{E}}}^{(0)} = \overrightarrow{\boldsymbol{\mathcal{J}}}^{(0)} = \boldsymbol{0}$, $\gamma = 1.9$, $\mu^{(0)} = 0.1$, $\mu_{\max} = 10^{10}$, $\varepsilon = 10^{-6}$, and $\zeta = 0$.

1: **while** not converged **do**
2: $\quad \overrightarrow{\boldsymbol{\mathcal{M}}}^{(\zeta+1)} = P_{\boldsymbol{\Phi}}(\frac{\boldsymbol{\mathcal{L}}*\overrightarrow{\boldsymbol{\mathcal{R}}}^{(\zeta)}+\overrightarrow{\boldsymbol{\mathcal{E}}}^{(\zeta)}+\mu\overrightarrow{\boldsymbol{\mathcal{Z}}}-\overrightarrow{\boldsymbol{\mathcal{J}}}^{(\zeta)}}{\mu+1}) + P_{\boldsymbol{\Phi}^c}(\boldsymbol{\mathcal{L}} * \overrightarrow{\boldsymbol{\mathcal{R}}}^{(\zeta)} + \overrightarrow{\boldsymbol{\mathcal{E}}}^{(\zeta)})$.
3: $\quad$ Update $\{\overrightarrow{\boldsymbol{\mathcal{R}}}^{(\zeta+1)}, \overrightarrow{\boldsymbol{\mathcal{E}}}^{(\zeta+1)}\}$ using Algorithm 4.
4: $\quad \overrightarrow{\boldsymbol{\mathcal{J}}}^{(\zeta+1)} = \overrightarrow{\boldsymbol{\mathcal{J}}}^{(\zeta)} + \mu^{(\zeta)}(P_{\boldsymbol{\Phi}}(\overrightarrow{\boldsymbol{\mathcal{M}}}^{(\zeta+1)}) - P_{\boldsymbol{\Phi}}(\overrightarrow{\boldsymbol{\mathcal{Z}}}))$.
5: $\quad \mu^{(\zeta+1)} = \min(\gamma\mu^{(\zeta)}, \mu_{\max})$.
6: $\quad$ Check the convergence conditions:
7: $\quad\quad \max(\|P_{\boldsymbol{\Phi}}(\overrightarrow{\boldsymbol{\mathcal{M}}}^{(\zeta+1)}) - P_{\boldsymbol{\Phi}}(\overrightarrow{\boldsymbol{\mathcal{Z}}})\|_F, \|\overrightarrow{\boldsymbol{\mathcal{R}}}^{(\zeta+1)} - \overrightarrow{\boldsymbol{\mathcal{R}}}^{(\zeta)}\|_F, \|\overrightarrow{\boldsymbol{\mathcal{E}}}^{(\zeta+1)} - \overrightarrow{\boldsymbol{\mathcal{E}}}^{(\zeta)}\|_F)/(n_1 n_3) < \varepsilon$.
8: $\quad \zeta = \zeta + 1$.
9: **end while**

**Output:** Optimal $\overrightarrow{\boldsymbol{\mathcal{M}}}^{\star} = \overrightarrow{\boldsymbol{\mathcal{M}}}^{(\zeta)}$, $\overrightarrow{\boldsymbol{\mathcal{R}}}^{\star} = \overrightarrow{\boldsymbol{\mathcal{R}}}^{(\zeta)}$ and $\overrightarrow{\boldsymbol{\mathcal{E}}}^{\star} = \overrightarrow{\boldsymbol{\mathcal{E}}}^{(\zeta)}$.

---

minimization strategy. At the $t$-th iteration, when $\boldsymbol{\mathcal{L}}_{t-1}$ is given, the update for $\{\overrightarrow{\boldsymbol{\mathcal{M}}}_t, \overrightarrow{\boldsymbol{\mathcal{R}}}_t, \overrightarrow{\boldsymbol{\mathcal{E}}}_t\}$ corresponds to solving

$$\{\overrightarrow{\boldsymbol{\mathcal{M}}}_t^{\star}, \overrightarrow{\boldsymbol{\mathcal{R}}}_t^{\star}, \overrightarrow{\boldsymbol{\mathcal{E}}}_t^{\star}\} = \underset{\overrightarrow{\boldsymbol{\mathcal{M}}}, \overrightarrow{\boldsymbol{\mathcal{R}}} \in \mathscr{P}, \overrightarrow{\boldsymbol{\mathcal{E}}}}{\arg\min} \frac{1}{2}\|\overrightarrow{\boldsymbol{\mathcal{M}}} - \boldsymbol{\mathcal{L}}_{t-1} * \overrightarrow{\boldsymbol{\mathcal{R}}} - \overrightarrow{\boldsymbol{\mathcal{E}}}\|_F^2 + \lambda_2\|\overrightarrow{\boldsymbol{\mathcal{E}}}\|_1 \quad \text{s.t. } P_{\boldsymbol{\Omega}_t}(\overrightarrow{\boldsymbol{\mathcal{M}}}) = P_{\boldsymbol{\Omega}_t}(\overrightarrow{\boldsymbol{\mathcal{Z}}}_t), \quad (16)$$

where $\boldsymbol{\Omega}_t = \{(i,k)|(i,t,k) \in \boldsymbol{\Omega}\}$. This problem can now be solved by using the Alternating Direction Method of Multipliers (ADMM) (Boyd et al., 2011). Specifically, the augmented Lagrangian function of (16) is

$$\hat{\ell}(\overrightarrow{\boldsymbol{\mathcal{M}}}, \overrightarrow{\boldsymbol{\mathcal{D}}}, \overrightarrow{\boldsymbol{\mathcal{R}}}, \overrightarrow{\boldsymbol{\mathcal{E}}}) = \frac{1}{2}\|\overrightarrow{\boldsymbol{\mathcal{M}}} - \boldsymbol{\mathcal{L}}_{t-1} * \overrightarrow{\boldsymbol{\mathcal{R}}} - \overrightarrow{\boldsymbol{\mathcal{E}}}\|_F^2 + \lambda_2\|\overrightarrow{\boldsymbol{\mathcal{E}}}\|_1$$
$$+ \langle \overrightarrow{\boldsymbol{\mathcal{J}}}, P_{\boldsymbol{\Omega}_t}(\overrightarrow{\boldsymbol{\mathcal{M}}}) - P_{\boldsymbol{\Omega}_t}(\overrightarrow{\boldsymbol{\mathcal{Z}}}_t)\rangle + \frac{\mu}{2}\|P_{\boldsymbol{\Omega}_t}(\overrightarrow{\boldsymbol{\mathcal{M}}}) - P_{\boldsymbol{\Omega}_t}(\overrightarrow{\boldsymbol{\mathcal{Z}}}_t)\|_F^2, \quad (17)$$

where $\overrightarrow{\boldsymbol{\mathcal{J}}}$ is the Lagrange multiplier and $\mu > 0$ is a penalty parameter. The implementation of the ADMM algorithm is outlined in Algorithm 3. Finally, we define $\boldsymbol{\mathcal{B}}_t = \sum_{i=1}^{t}(\overrightarrow{\boldsymbol{\mathcal{M}}}_i^{\star} - \overrightarrow{\boldsymbol{\mathcal{E}}}_i^{\star}) * \overrightarrow{\boldsymbol{\mathcal{R}}}_i^{\star T}$ and the update of $\boldsymbol{\mathcal{L}}_t$ is exactly the same as in OMRTD. We dub this approach *robust OMRTD* (rOMRTD) in our experiments.

## 5 Experiments

In this section, we present several experimental results on both synthetic and real data. All experiments are conducted on a PC with an AMD Ryzen 9 5950X 3.40GHz CPU and 64GB RAM with Matlab R2023b. We set $\lambda_1 = \lambda_2 = 1/\sqrt{n_1}$ for OMRTD/rOMRTD, and we follow the default parameter settings for the baselines.

### 5.1 Synthetic Data Experiments

**Data generation** We generate the clean data tensor $\boldsymbol{\mathcal{X}} = \boldsymbol{\mathcal{U}} * \boldsymbol{\mathcal{V}}^T$, where the entries of $\boldsymbol{\mathcal{U}} \in \mathbb{R}^{n_1 \times r \times n_3}$ and $\boldsymbol{\mathcal{V}} \in \mathbb{R}^{N \times r \times n_3}$ are drawn i.i.d. from $\mathcal{N}(0, 1)$ distribution. Here, we set $n_1 = 50$ and $n_3 = 20$. The observed data tensor $\boldsymbol{\mathcal{Z}}$ is generated by $\boldsymbol{\mathcal{Z}} = \boldsymbol{\mathcal{X}} + \boldsymbol{\mathcal{E}}$, where $\boldsymbol{\mathcal{E}}$ is a sparse tensor with a fraction of $\rho$ non-zero entries. The elements in $\boldsymbol{\mathcal{E}}$ are from a uniform distribution over the interval of $[-10, 10]$.

**Evaluation metric** We evaluate the fitness of the recovered tensor subspace $\boldsymbol{\mathcal{L}}$ (with each frontal slice being normalized) and the ground truth $\boldsymbol{\mathcal{U}}$ based on the idea of Expressed Variance (EV) (Xu et al., 2010):

$$\text{EV}(\boldsymbol{\mathcal{U}}; \boldsymbol{\mathcal{L}}) = \frac{\|\boldsymbol{\mathcal{U}}^T * \boldsymbol{\mathcal{L}}\|_F^2}{\|\boldsymbol{\mathcal{U}}\|_F^2} = \frac{\text{tr}((\boldsymbol{\mathcal{U}} * \boldsymbol{\mathcal{U}}^T * \boldsymbol{\mathcal{L}} * \boldsymbol{\mathcal{L}}^T)(:,:,1))}{\text{tr}((\boldsymbol{\mathcal{U}}^T * \boldsymbol{\mathcal{U}})(:,:,1))}.$$

The value of EV ranges between 0 and 1 and a higher value indicates better recovery. The Monte Carlo simulations are repeated 10 times and we report the averaged EV of these 10 random trials.

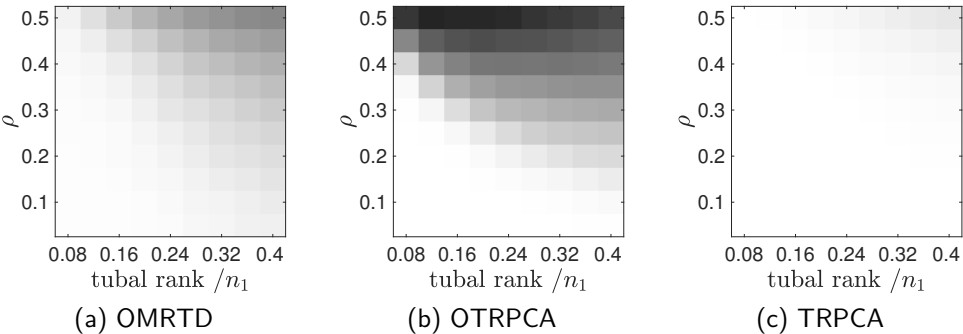

Figure 1: Performance of tensor subspace recovery using complete data under different intrinsic dimensions and corruptions. Brighter cells represent better performance.

**Baselines** For the complete data experiments, we compare the performance of OMRTD with OTRPCA (Zhang et al., 2016) and TRPCA (Lu et al., 2020). For the case of data having missing entries, we choose an approach that first performs tensor completion based on the tensor nuclear norm (TNN) (Zhang & Aeron, 2017; Lu et al., 2018) and then conducts TRPCA (Lu et al., 2020) on the recovered data (TNN+TRPCA) as the baseline.

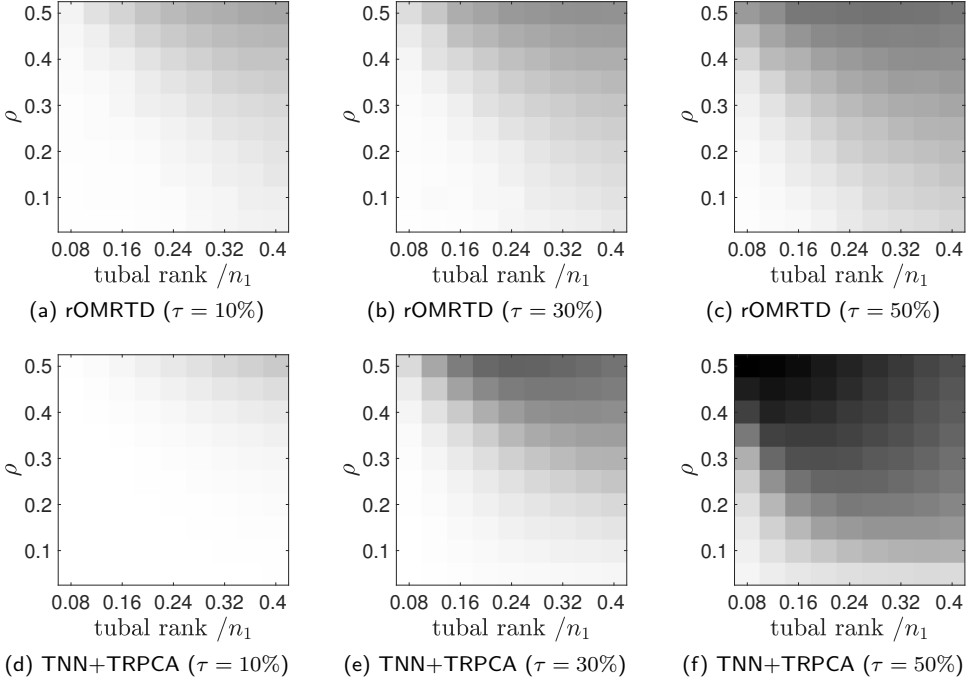

Figure 2: Performance of tensor subspace recovery using missing data under different intrinsic dimensions and corruptions. Brighter cells represent better performance.

### 5.1.1 Robustness

We first study the robustness of OMRTD in terms of EV value, and compare it to the tensor nuclear norm based OTRPCA and the batch algorithm TRPCA. In this set of experiments, the total number of samples $N = 2000$. We vary the true tubal rank from $0.08n_1$ to $0.4n_1$, with a step size $0.04n_1$, and the corruption fraction $\rho$ ranges from 0.05 to 0.5, with a step size 0.05. The results are represented in Figure 1. Since

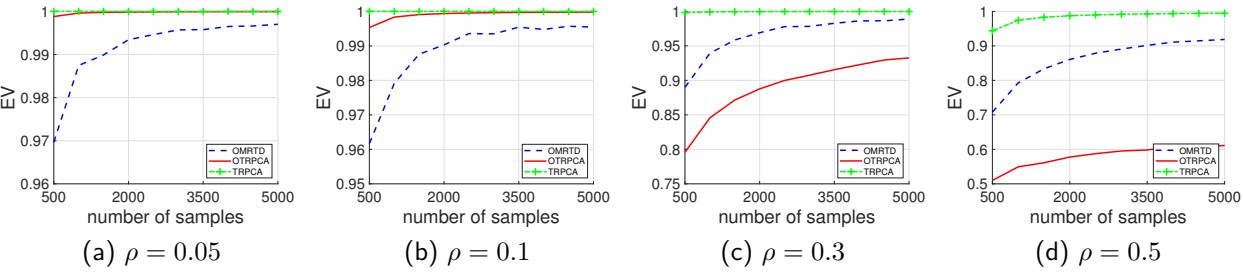

Figure 3: EV value against the number of samples under different corruption fractions.

TRPCA accesses all the data in each iteration, it always achieves the best performance. We observe that both OMRTD and OTRPCA perform comparably in easy settings (i.e., few corruption and low tubal rank). However, in tough cases, OMRTD outperforms OTRPCA. For example, when the true tubal rank is 12 and $\rho = 0.5$, the EV values of OMRTD and OTRPCA are 0.8274 and 0.5840, respectively. In order to further investigate this phenomenon, we plot the EV curve against the fraction of corruption for some given tubal ranks in Figure 5 in Appendix A.3. Notably, when manipulating a low tubal rank tensor, OTRPCA exhibits similar performance compared to OMRTD under a low level of noise, e.g., the true tubal rank is no more than 8 and $\rho$ is no more than 0.25. However, as the true tubal rank gets larger or the fraction of corruption increases, OTRPCA degrades faster than OMRTD. This is possibly because the proposed tensor max-norm is a tighter approximation to the tensor tubal rank.

Next, we study the effectiveness of rOMRTD in the missing data scenario, where we set the percentage of missing entries $\tau$ to be 10%, 30% and 50%. Figure 2 indicates that TNN+TRPCA performs better than rOMRTD when $\tau = 10\%$. When the number of missing entries increases, the EV values of TNN+TRPCA drop rapidly, especially when the fraction of corruption becomes large. Finally, rOMRTD outperforms TNN+TRPCA for almost all different tubal ranks and $\rho$'s when $\tau = 50\%$. The detailed plot of the EV curve against the fraction of corruption under some specific tubal ranks for $\tau = 30\%$ is shown in Figure 6 in Appendix A.3.

### 5.1.2 Convergence Rate

We now examine the convergence of OMRTD in terms of the EV curve as a function of the number of samples. We first fix the true tubal rank to be $0.2n_1 = 10$. The results are depicted in Figure 3. As expected, TRPCA achieves the best performance since it is a batch method and it requires to access all the data during optimization. OMRTD is comparable to OTRPCA when the corruption level is low (see Figure 3a and Figure 3b) and the gap between the EV values for these two methods is below 0.04. When data are grossly corrupted, OMRTD converges faster than OTRPCA (see Figure 3c and Figure 3d), which again suggests that tensor max-norm might be a better fit than the tensor nuclear norm when the signal to noise ratio is low.

We then compare the convergence rate of OMRTD and OTRPCA under different $n_1$'s in Figure 4a and Figure 4b. The tubal rank of data is set to be $0.1n_1$ and the error corruption $\rho$ is fixed to be 0.3. We observe that when $n_1 = 50$, OMRTD is generally slightly worse than OTRPCA and the gap between the EV values is below 0.004 for the same number of samples. However, when $n_1 = 100$, OMRTD significantly outperforms OTRPCA. It attains the EV value of 0.95 only with accessing 1000 samples, whereas OTRPCA cannot obtain the same accuracy even using 20000 samples.

### 5.1.3 Computational Complexity

As we discussed, when we solve the dual problem to optimize $\overrightarrow{\mathcal{R}}$, the initial solution $\overrightarrow{\mathcal{R}}_{\text{cand}}$ may violate the constraint. Thus, OMRTD is inferior to OTRPCA in terms of computation. We plot the running time with respect to the number of samples for $n_1 \in \{50, 100\}$ in Figure 4c and Figure 4d, which show that OTRPCA is about 2.4 times faster than OMRTD. When $n_1 = 50$, OMRTD and OTRPCA take 76 seconds to achieve

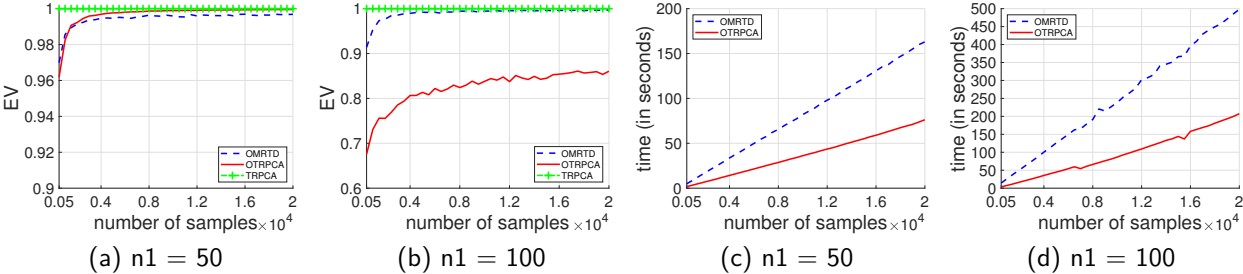

Figure 4: (a)-(b) EV value as a function of the number of samples for $n_1 \in \{50, 100\}$. The intrinsic dimension is $0.1n_1$ and the corruption fraction $\rho = 0.3$. (c)-(d) Running time as a function of the number of samples for $n_1 \in \{50, 100\}$.

Table 1: Background subtraction results and algorithm running time (in seconds) on CAMO-UOW dataset.

| Methods | Metrics | Video 1 | Video 2 | Video 3 | Video 4 | Video 5 | Video 6 | Video 7 | Video 8 | Video 9 | Video 10 |
|---------|---------|---------|---------|---------|---------|---------|---------|---------|---------|---------|----------|
| RPCA | Precision | 0.7290 | 0.7873 | 0.6367 | **0.7698** | 0.6472 | **0.9883** | **0.9521** | 0.7343 | 0.7552 | **0.9646** |
| | Recall | 0.2032 | 0.2237 | 0.2328 | 0.1903 | 0.1174 | **0.7209** | 0.1621 | 0.2334 | 0.1101 | **0.4262** |
| | F-measure | 0.3038 | 0.3311 | 0.3337 | 0.2982 | 0.1941 | **0.8288** | 0.2608 | 0.3476 | 0.1628 | **0.5557** |
| | Time | 264.1 | 130.1 | 255.5 | 258.9 | 265.5 | 247.3 | 207.9 | 359 | 265.3 | 382.2 |
| TRPCA | Precision | 0.5829 | 0.6281 | 0.5712 | 0.6632 | 0.6665 | 0.9541 | 0.7954 | 0.6021 | 0.7197 | 0.8734 |
| | Recall | 0.1785 | 0.1019 | 0.1045 | 0.0926 | 0.0861 | 0.3865 | 0.0826 | 0.1037 | 0.0530 | 0.1786 |
| | F-measure | 0.2599 | 0.1489 | 0.1673 | 0.1613 | 0.1483 | 0.5411 | 0.1460 | 0.1635 | 0.0949 | 0.2761 |
| | Time | 1086 | 518.4 | 1111 | 1102 | 1121 | 1103 | 933.8 | 1526 | 979.9 | 1490 |
| ORPCA | Precision | 0.0435 | 0.1336 | 0.1711 | 0.0602 | 0.0404 | 0.1339 | 0.083 | 0.1226 | 0.0372 | 0.1528 |
| | Recall | 0.1923 | 0.2859 | **0.4174** | 0.1990 | 0.1278 | 0.7105 | 0.2071 | 0.3010 | 0.1231 | 0.4325 |
| | F-measure | 0.0669 | 0.1715 | 0.2311 | 0.0910 | 0.0593 | 0.2160 | 0.1118 | 0.1657 | 0.0536 | 0.2165 |
| | Time | 23.13 | 12.17 | 20.75 | 21.51 | 30.99 | 19.32 | 11.47 | 54.31 | 20.63 | 21.73 |
| OMRMD | Precision | 0.6806 | 0.7073 | 0.6510 | 0.6162 | 0.6306 | 0.9038 | 0.6285 | 0.7052 | 0.6573 | 0.7801 |
| | Recall | 0.1777 | 0.0888 | 0.1016 | 0.1184 | 0.0921 | 0.3038 | 0.1149 | 0.1239 | 0.0876 | 0.2070 |
| | F-measure | 0.2657 | 0.1489 | 0.1718 | 0.1965 | 0.1500 | 0.4492 | 0.1798 | 0.1993 | 0.1437 | 0.3123 |
| | Time | **8.74** | **4.28** | **8.73** | **8.73** | **8.90** | **8.88** | **6.91** | **11.76** | **7.49** | **11.56** |
| OTRPCA | Precision | 0.3580 | 0.4024 | 0.4636 | 0.5077 | 0.4260 | 0.9475 | 0.7607 | 0.5560 | 0.5543 | 0.8462 |
| | Recall | 0.1595 | 0.0923 | 0.2526 | 0.1171 | 0.0693 | 0.5023 | 0.0575 | 0.1208 | 0.0561 | 0.2809 |
| | F-measure | 0.2095 | 0.1328 | 0.3209 | 0.1862 | 0.1007 | 0.6493 | 0.1056 | 0.1853 | 0.0849 | 0.3991 |
| | Time | 31.67 | 14.70 | 31.65 | 31.82 | 31.87 | 31.89 | 26.43 | 44.96 | 27.99 | 44.27 |
| OMRTD | Precision | **0.8663** | **0.8682** | **0.9638** | 0.7617 | **0.8200** | 0.9382 | 0.7992 | **0.8597** | **0.8221** | 0.8544 |
| | Recall | **0.3769** | **0.3243** | 0.2573 | **0.2447** | **0.2446** | 0.4555 | **0.2823** | **0.3454** | **0.1925** | 0.3220 |
| | F-measure | **0.5119** | **0.4657** | **0.3861** | **0.3624** | **0.3668** | 0.6103 | **0.4080** | **0.4830** | **0.2901** | 0.4552 |
| | Time | 83.01 | 39.02 | 82.61 | 79.1 | 84.42 | 84.55 | 67.95 | 119.7 | 74.44 | 132.6 |

the EV values of 0.9963 (with 9500 samples) and 0.9995 (with 20000 samples), respectively. The caveat here is that when $n_1 = 100$, OMRTD and OTRPCA take 207 seconds to achieve the EV values of around 0.9945 (with a little bit more than 8000 samples) and 0.86 (with 20000 samples), respectively. We can even expect that the gap between the EV values of these two methods will get even larger as $n_1$ increases. It is reasonable to conclude that the advantage of OMRTD over OTRPCA in terms of convergence rate significantly outweighs the increase in computational complexity when $n_1$ is moderately large.

## 5.2 Real Data Experiments

In this subsection, we compare the performance of OMRTD with OTRPCA (Zhang et al., 2016), TRPCA (Lu et al., 2020), OMRMD (Shen et al., 2017), ORPCA (Feng et al., 2013) and robust PCA (RPCA) (Candès et al., 2011) on the CAMO-UOW dataset (Li et al., 2017) for video background subtraction. The task is to separate the moving foreground objects, which are usually sparsely distributed in the video frames, from a

static background, which can be characterized by a low-rank matrix/tensor. The dataset contains 10 real video sequences and we use all these sequences for both qualitative and quantitative analysis. To evaluate the performance of OMRTD, the Precision, Recall, and F-measure, are used as basic evaluation metrics. The upper bound of the rank of the clean data in OMRMD/ORPCA is set to be 5 and the upper bound of the tubal rank of the clean tensor in OMRTD/OTRPCA is set to be 3. As can be seen from Table 1, OMRTD achieves the highest F-measure scores for 8 videos. The visual comparison in Figure 7 in Appendix A.3 shows that our method is competent to extract the foreground from these videos.

## 6  Conclusion

In this paper, we have proposed a tensor max-norm for low-rank tensor modeling and developed an online algorithm for the max-norm regularized tensor decomposition (OMRTD) problem. The main idea of OM-RTD is to reformulate the objective function of max-norm regularized tensor decomposition as a constrained problem using the tensor factorization form of the max-norm, which can be solved by stochastic optimization. We further extended the proposed method to the missing data scenario. Comprehensive simulations demonstrate the effectiveness of OMRTD and we conjecture that the tensor max-norm might be a tighter relaxation of the tensor average rank compared to the tensor nuclear norm.

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

## A  Appendix

### A.1  Some Proofs

We first present the proof of Proposition 1.

*Proof.* Let us denote $\kappa = \|\bar{\mathbf{R}}\|_{2,\infty}$, which is positive as long as $\mathcal{R}$ is not a zero tensor. Otherwise the recovered low tubal-rank component $\mathcal{X} = \mathcal{L} * \mathcal{R}^T$ is a zero tensor. Now we construct two variables $\widetilde{\mathcal{L}} = \kappa\mathcal{L}$ and $\widetilde{\mathcal{R}} = \frac{1}{\kappa}\mathcal{R}$, and replace $\mathcal{L}$ and $\mathcal{R}$ in (5) with $\frac{1}{\kappa}\widetilde{\mathcal{L}}$ and $\kappa\widetilde{\mathcal{R}}$, respectively. Our problem can be written as

$$\min_{\widetilde{\mathcal{L}},\widetilde{\mathcal{R}},\mathcal{E}} \frac{1}{2}\|\mathcal{Z} - (\frac{1}{\kappa}\widetilde{\mathcal{L}}) * (\kappa\widetilde{\mathcal{R}})^T - \mathcal{E}\|_F^2 + \frac{\lambda_1}{2}\|\frac{1}{\kappa}\bar{\widetilde{\mathbf{L}}}\|_{2,\infty}^2\|\kappa\bar{\widetilde{\mathbf{R}}}\|_{2,\infty}^2 + \lambda_2\|\mathcal{E}\|_1,$$

which is equivalent to solving

$$\min_{\widetilde{\mathcal{L}},\widetilde{\mathcal{R}},\mathcal{E}} \frac{1}{2}\|\mathcal{Z} - \widetilde{\mathcal{L}} * \widetilde{\mathcal{R}}^T - \mathcal{E}\|_F^2 + \frac{\lambda_1}{2}\|\bar{\widetilde{\mathbf{L}}}\|_{2,\infty}^2\|\bar{\widetilde{\mathbf{R}}}\|_{2,\infty}^2 + \lambda_2\|\mathcal{E}\|_1.$$

Since $\widetilde{\mathcal{R}} = \frac{1}{\kappa}\mathcal{R}$, we have $\|\bar{\widetilde{\mathbf{R}}}\|_{2,\infty} = \frac{1}{\kappa}\|\bar{\mathbf{R}}\|_{2,\infty} = 1$. Therefore, we can reformulate our MRTD problem as a constrained program:

$$\min_{\widetilde{\mathcal{L}},\widetilde{\mathcal{R}},\mathcal{E}} \frac{1}{2}\|\mathcal{Z} - \widetilde{\mathcal{L}} * \widetilde{\mathcal{R}}^T - \mathcal{E}\|_F^2 + \frac{\lambda_1}{2}\|\bar{\widetilde{\mathbf{L}}}\|_{2,\infty}^2 + \lambda_2\|\mathcal{E}\|_1 \quad \text{s.t.} \quad \|\bar{\widetilde{\mathbf{R}}}\|_{2,\infty}^2 = 1.$$

To see why the above problem is equivalent to (6), we only need to show that any optimal solution $(\mathcal{L}^\star, \mathcal{R}^\star, \mathcal{E}^\star)$ to (6) must satisfy $\|\bar{\mathbf{R}}^\star\|_{2,\infty} = 1$. Again, suppose that $\kappa = \|\bar{\mathbf{R}}^\star\|_{2,\infty} < 1$. Let $\mathcal{L}' = \kappa\mathcal{L}^\star$ and $\mathcal{R}' = \frac{1}{\kappa}\mathcal{R}^\star$. It is clear that $(\mathcal{L}', \mathcal{R}', \mathcal{E}^\star)$ are still feasible. However, the objective value now becomes

$$\frac{1}{2}\|\mathcal{Z} - \mathcal{L}' * \mathcal{R}'^T - \mathcal{E}^\star\|_F^2 + \frac{\lambda_1}{2}\|\bar{\mathbf{L}}'\|_{2,\infty}^2 + \lambda_2\|\mathcal{E}^\star\|_1$$
$$= \frac{1}{2}\|\mathcal{Z} - \mathcal{L}^\star * \mathcal{R}^{\star T} - \mathcal{E}^\star\|_F^2 + \frac{\lambda_1}{2}\kappa^2\|\bar{\mathbf{L}}^\star\|_{2,\infty}^2 + \lambda_2\|\mathcal{E}^\star\|_1$$
$$< \frac{1}{2}\|\mathcal{Z} - \mathcal{L}^\star * \mathcal{R}^{\star T} - \mathcal{E}^\star\|_F^2 + \frac{\lambda_1}{2}\|\bar{\mathbf{L}}^\star\|_{2,\infty}^2 + \lambda_2\|\mathcal{E}^\star\|_1,$$

which contradicts the assumption that $(\mathcal{L}^\star, \mathcal{R}^\star, \mathcal{E}^\star)$ is optimal. Thus we complete the proof. $\square$

Next, we prove Proposition 2.

*Proof.* (Sketch) Let us consider the optimization problem (6). $\overrightarrow{\mathcal{R}}_t^\star$ is uniformly bounded because of the constraint that $\|\bar{\mathbf{R}}\|_{2,\infty}^2 \leq 1$. Plugging in the trivial solution $\{\overrightarrow{\mathcal{R}}_t', \overrightarrow{\mathcal{E}}_t'\} = \{\mathbf{0}, \mathbf{0}\}$, we have $\widetilde{\ell}(\overrightarrow{\mathcal{Z}}_t, \mathcal{L}_{t-1}, \mathbf{0}, \mathbf{0}) = \frac{1}{2}\|\overrightarrow{\mathcal{Z}}_t\|_F^2$. Therefore the optimal solution should satisfy

$$\frac{1}{2}\|\overrightarrow{\mathcal{Z}}_t - \mathcal{L}_{t-1} * \overrightarrow{\mathcal{R}}_t^\star - \overrightarrow{\mathcal{E}}_t^\star\|_F^2 + \lambda_2\|\overrightarrow{\mathcal{E}}_t^\star\|_1 \leq \frac{1}{2}\|\overrightarrow{\mathcal{Z}}_t\|_F^2,$$

which implies $\|\overrightarrow{\boldsymbol{\mathcal{E}}}_t^\star\|_1 \leq \frac{1}{2\lambda_2}\|\overrightarrow{\boldsymbol{\mathcal{Z}}}_t\|_F^2$. Since $\overrightarrow{\boldsymbol{\mathcal{Z}}}_t$ is uniformly bounded (Assumption (A1)), $\overrightarrow{\boldsymbol{\mathcal{E}}}_t^\star$ is uniformly bounded. The uniform boundedness of $\frac{1}{t}\boldsymbol{\mathcal{A}}_t$ and $\frac{1}{t}\boldsymbol{\mathcal{B}}_t$ follows immediately. Recall that $\bar{\mathbf{L}}_t$ is the optimal basis for (14). Thus, the subgradient of the objective function with respect to $\bar{\mathbf{L}}_t$ should contain zero, i.e., $\frac{1}{t}\bar{\mathbf{L}}_t\bar{\mathbf{A}}_t - \frac{1}{t}\bar{\mathbf{B}}_t + \frac{\lambda_1 n_3}{t}\mathbf{H}_t = \mathbf{0}$, where $\mathbf{H}_t$ is the subgradient of $\frac{1}{2}\|\bar{\mathbf{L}}_t\|_{2,\infty}^2$. Since all of the eigenvalues of $\frac{1}{t}\bar{\mathbf{A}}_t$ are lower bounded by a positive constant (Assumption (A2)), $\frac{1}{t}\bar{\mathbf{A}}_t$ is invertible. Thus, $\bar{\mathbf{L}}_t = (\frac{1}{t}\bar{\mathbf{B}}_t - \frac{\lambda_1 n_3}{t}\mathbf{H}_t)(\frac{1}{t}\bar{\mathbf{A}}_t)^{-1}$. Following the proof of Shen et al. (2017, Proposition 7), $\bar{\mathbf{L}}_t$, and equivalently, $\boldsymbol{\mathcal{L}}_t$, can be uniformly bounded. □

We now prove Corollary 1 as follows.

*Proof.* Since $\overrightarrow{\boldsymbol{\mathcal{Z}}}_t$, $\boldsymbol{\mathcal{L}}_t$, $\overrightarrow{\boldsymbol{\mathcal{R}}}_t^\star$ and $\overrightarrow{\boldsymbol{\mathcal{E}}}_t^\star$ are all uniformly bounded, it is easy to show $\widetilde{\ell}(\overrightarrow{\boldsymbol{\mathcal{Z}}}_t, \boldsymbol{\mathcal{L}}_t, \overrightarrow{\boldsymbol{\mathcal{R}}}_t^\star, \overrightarrow{\boldsymbol{\mathcal{E}}}_t^\star)$ and $\ell(\boldsymbol{\mathcal{L}}_t, \overrightarrow{\boldsymbol{\mathcal{Z}}}_t)$ are uniformly bounded from above. Thus, $g_t(\boldsymbol{\mathcal{L}}_t)$ is also bounded. To show that $g_t(\boldsymbol{\mathcal{L}})$ is uniformly Lipschitz, we first define $\bar{g}_t(\bar{\mathbf{L}}) = g_t(\boldsymbol{\mathcal{L}}) = \frac{1}{2tn_3}\|\bar{\mathbf{Z}}_t - \bar{\mathbf{L}}\bar{\mathbf{R}}_t^\star - \bar{\mathbf{E}}_t^\star\|_F^2 + \frac{1}{t}\sum_{i=1}^t \lambda_2\|\overrightarrow{\boldsymbol{\mathcal{E}}}_t^\star\|_1 + \frac{\lambda_1}{2t}\|\bar{\mathbf{L}}\|_{2,\infty}^2$, then the subgradient of $\bar{g}_t$ with respect to $\bar{\mathbf{L}}$ is

$$\|\nabla_{\bar{\mathbf{L}}}\bar{g}_t(\bar{\mathbf{L}})\|_F = \|\frac{1}{tn_3}(\bar{\mathbf{L}}\bar{\mathbf{A}}_t - \bar{\mathbf{B}}_t) + \frac{\lambda_1}{t}\mathbf{H}\|_F \leq \|\frac{1}{tn_3}(\bar{\mathbf{L}}\bar{\mathbf{A}}_t - \bar{\mathbf{B}}_t)\|_F + \lambda_1\|\bar{\mathbf{L}}\|_F,$$

where $\mathbf{H}$ is the subgradient of $\frac{1}{2}\|\bar{\mathbf{L}}\|_{2,\infty}^2$. Since $\bar{\mathbf{L}}$, $\frac{1}{t}\bar{\mathbf{A}}_t$ and $\frac{1}{t}\bar{\mathbf{B}}_t$ are all uniformly bounded, the subgradient of $\bar{g}_t$ is uniformly bounded. Notice that each entry of $\bar{\boldsymbol{\mathcal{L}}}$ (in other words, each entry on the main diagonal block of $\bar{\mathbf{L}}$) is a linear combination of the entries in the same mode-3 fiber of $\boldsymbol{\mathcal{L}}$, where the corresponding coefficients are the elements of the Discrete Fourier transform matrix. Thus the subgradient of $g_t(\boldsymbol{\mathcal{L}})$ is also uniformly bounded for all $\boldsymbol{\mathcal{L}} \in \mathscr{L}$ and $g_t(\boldsymbol{\mathcal{L}})$ is Lipschitz. □

## A.2 Algorithm Details

---
**Algorithm 4** Data Projection (Problem (10))
---
**Input:** Observed data sample $\overrightarrow{\boldsymbol{\mathcal{Z}}} \in \mathbb{R}^{n_1 \times 1 \times n_3}$, $\boldsymbol{\mathcal{L}} \in \mathbb{R}^{n_1 \times r \times n_3}$, and parameters $\lambda_2$ and $\epsilon$.
**Initialize:** $\overrightarrow{\boldsymbol{\mathcal{E}}} = \mathbf{0}$.
1: **while** not converged **do**
2:     Compute the potential solution $\overrightarrow{\boldsymbol{\mathcal{R}}}_{\text{cand}}$ using (11).
3:     **if** $\forall k, \|\overrightarrow{\mathbf{R}}_{\text{cand}}^{(k)}\|_2 \leq 1$ **then**
4:         Set $\overrightarrow{\boldsymbol{\mathcal{R}}} = \overrightarrow{\boldsymbol{\mathcal{R}}}_{\text{cand}}$.
5:     **else**
6:         **for** $k = 1, 2, \ldots, n_3$ **do**
7:             **if** $\|\overrightarrow{\mathbf{R}}_{\text{cand}}^{(k)}\|_2 \leq 1$ **then**
8:                 Set $\overrightarrow{\mathbf{R}}^{(k)} = \overrightarrow{\mathbf{R}}_{\text{cand}}^{(k)}$.
9:             **else**
10:                Update $\overrightarrow{\mathbf{R}}^{(k)}$ by Algorithm 5.
11:             **end if**
12:         **end for**
13:     **end if**
14:     $\overrightarrow{\boldsymbol{\mathcal{R}}} = \texttt{ifft}(\overrightarrow{\boldsymbol{\mathcal{R}}}, [\,], 3)$.
15:     $\overrightarrow{\boldsymbol{\mathcal{E}}} = S_{\lambda_2}[\overrightarrow{\boldsymbol{\mathcal{Z}}} - \boldsymbol{\mathcal{L}} * \overrightarrow{\boldsymbol{\mathcal{R}}}]$.
16: **end while**
**Output:** Optimal $\overrightarrow{\boldsymbol{\mathcal{R}}}^\star$ and $\overrightarrow{\boldsymbol{\mathcal{E}}}^\star$.

---

Let $\{\overrightarrow{\boldsymbol{\mathcal{R}}}', \overrightarrow{\boldsymbol{\mathcal{E}}}'\}$ and $\{\overrightarrow{\boldsymbol{\mathcal{R}}}'', \overrightarrow{\boldsymbol{\mathcal{E}}}''\}$ be the two consecutive iterates. If $\max(\|\overrightarrow{\boldsymbol{\mathcal{R}}}' - \overrightarrow{\boldsymbol{\mathcal{R}}}''\|_F, \|\overrightarrow{\boldsymbol{\mathcal{E}}}' - \overrightarrow{\boldsymbol{\mathcal{E}}}''\|_F)/(n_1 n_3)$ is less than $10^{-6}$, or the number of iterations exceeds 100, we will terminate Algorithm 4.

---

**Algorithm 5** Bisection Method for Solving Problem (12)

---

**Input:** $\mathbf{L} \in \mathbb{C}^{n_1 \times r}$, $\mathbf{z} \in \mathbb{C}^{n_1}$, $\mathbf{e} \in \mathbb{C}^{n_1}$.

**Initialize:** Set $\eta_1 = 0$ and $\eta_2$ to be large enough such that $\|\overrightarrow{\overline{\mathbf{R}}}^{(k)}(\eta_2)\|_F \leq 1$.

1: **repeat**
2:      Compute the middle point: $\eta^{(k)} = \frac{1}{2}(\eta_1 + \eta_2)$.
3:      **if** $\|\overrightarrow{\overline{\mathbf{R}}}^{(k)}(\eta^{(k)})\|_F < 1$ **then**
4:          Update $\eta_2 = \eta^{(k)}$.
5:      **else**
6:          Update $\eta_1 = \eta^{(k)}$.
7:      **end if**
8: **until** $\|\overrightarrow{\overline{\mathbf{R}}}^{(k)}\|_F = 1$

**Output:** Optimal $\overrightarrow{\overline{\mathbf{R}}}^{(k)}$ and $\eta^{(k)}$.

---

**Algorithm 6** The Update of $\boldsymbol{\mathcal{L}}$

---

**Input:** $\boldsymbol{\mathcal{L}} \in \mathbb{R}^{n_1 \times r \times n_3}$ in the previous iteration, accumulation tensors $\boldsymbol{\mathcal{A}}$ and $\boldsymbol{\mathcal{B}}$, and parameter $\lambda_1$.

1: $\bar{\boldsymbol{\mathcal{L}}} = \mathtt{fft}(\boldsymbol{\mathcal{L}}, [\,], 3)$, $\bar{\boldsymbol{\mathcal{A}}} = \mathtt{fft}(\boldsymbol{\mathcal{A}}, [\,], 3)$, and $\bar{\boldsymbol{\mathcal{B}}} = \mathtt{fft}(\boldsymbol{\mathcal{B}}, [\,], 3)$.
2: Compute the subgradient of $\frac{1}{2}\|\bar{\mathbf{L}}\|_{2,\infty}^2$: $\mathbf{H} = \partial(\frac{1}{2}\|\bar{\mathbf{L}}\|_{2,\infty}^2)$.
3: **for** $k = 1, 2, \ldots, n_3$ **do**
4:      $\widetilde{\mathbf{A}} = \bar{\mathbf{A}}^{(k)}$, $\widetilde{\mathbf{B}} = \bar{\mathbf{B}}^{(k)}$, $\widetilde{\mathbf{H}} = \mathbf{H}_k$.
5:      **for** $j = 1, 2, \ldots, r$ **do**
6:          $\bar{\boldsymbol{\mathcal{L}}}(:, j, k) = \bar{\boldsymbol{\mathcal{L}}}(:, j, k) - \frac{1}{\widetilde{\mathbf{A}}_{j,j}}\left(\frac{1}{n_3}\left(\bar{\mathbf{L}}^{(k)}\widetilde{\mathbf{a}}_j - \widetilde{\mathbf{b}}_j\right) + \lambda_1\widetilde{\mathbf{h}}_j\right)$.
7:      **end for**
8: **end for**

**Output:** $\boldsymbol{\mathcal{L}} = \mathtt{ifft}(\bar{\boldsymbol{\mathcal{L}}}, [\,], 3)$.

---

For Algorithm 6, we find that a one-pass update on the dictionary $\mathcal{L}$ is sufficient to guarantee a desirable accuracy, as we shown in the experiments. This is also observed in Mairal et al. (2010).

### A.3 Supplementary Experimental Results

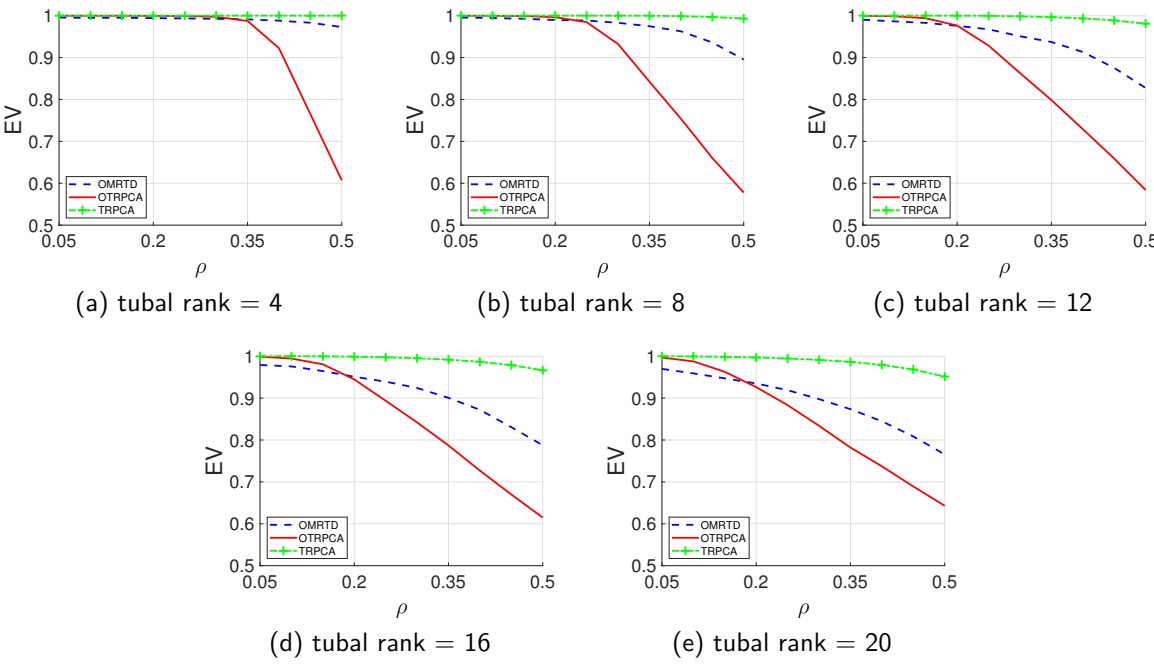

Figure 5: EV value as a function of corruption fraction for different intrinsic dimensions of complete data.

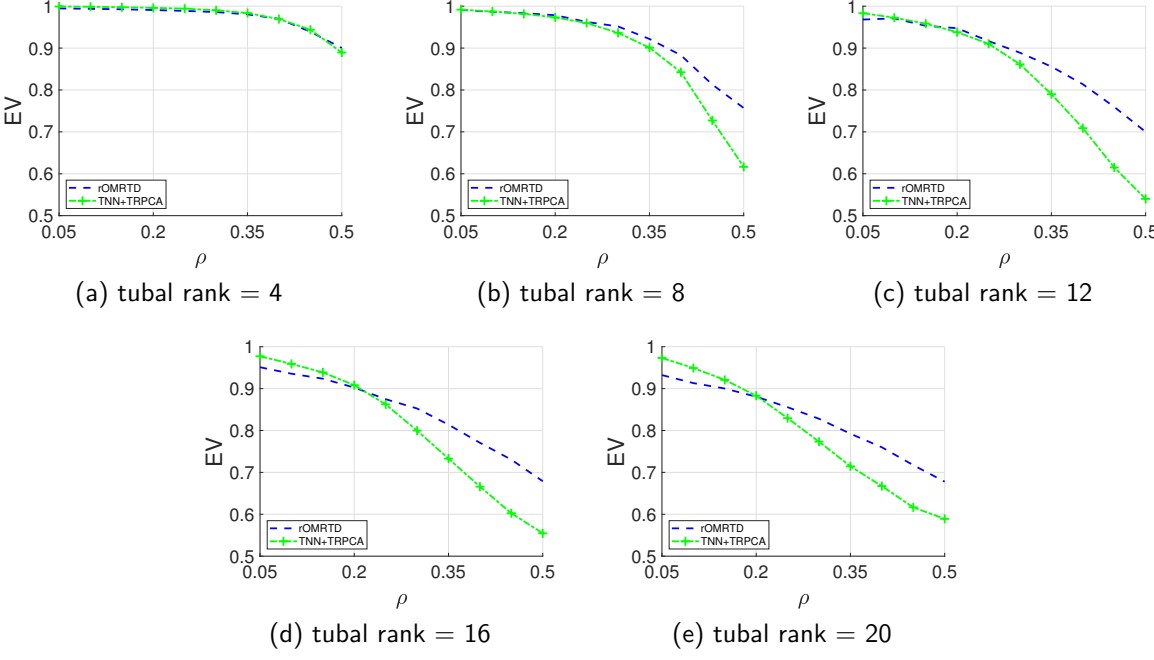

Figure 6: EV value as a function of corruption fraction for different intrinsic dimensions of missing data when $\tau = 30\%$.

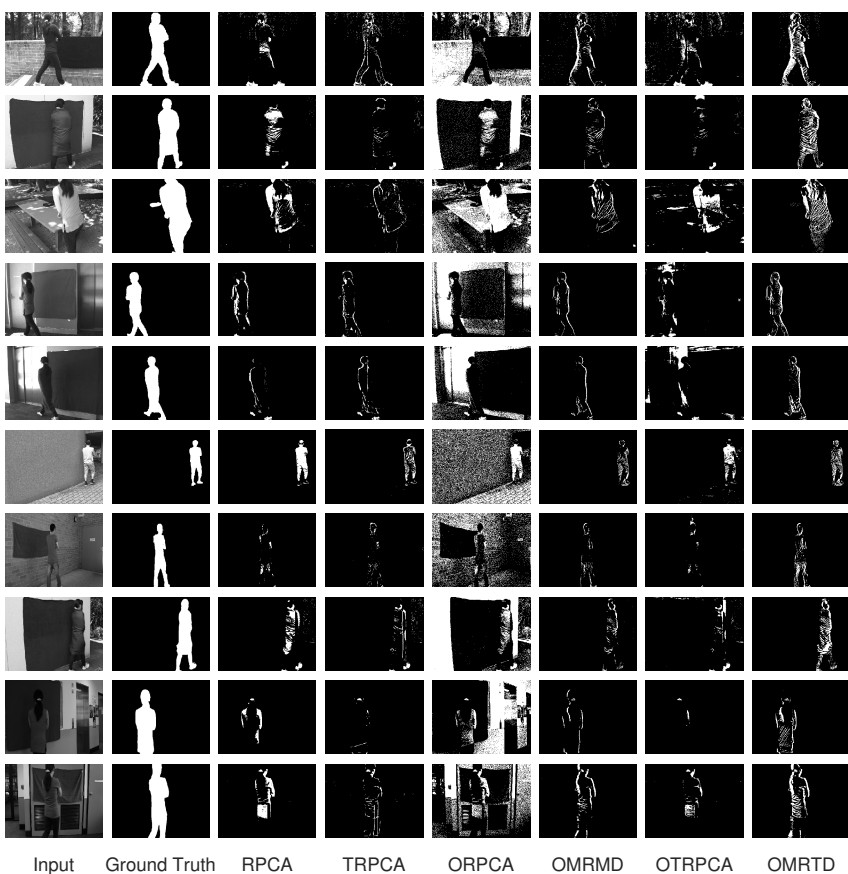

Figure 7: Examples of background subtraction using the CAMO-UOW dataset (Li et al., 2017). From top to bottom are 10 sequences within the dataset.

