# OpenReview forum: "Online Tensor Max-Norm Regularization via Stochastic Optimization"
_TMLR — Accepted by TMLR_

### Review · Reviewer_sMav · 2024-03-05

**Summary Of Contributions:**

The authors propose an online max-norm regularized tensor decomposition (OMRTD).  The proposed method learns a low-rank tensor factorization under the t-product with max-norm regularization. The authors formulate an alternating optimization problem slice-wise and add practical constraints to their low-rank factorization to make online updating a feasible strategy.  The authors describe convergence and complexity behavior of their algorithm and show numerical experiments on synthetic data for a tensor completion problem with comparisons to related algorithms in the literature.

Contributions:
- A new OMRTD algorithm that can handle larger-scale tensor completion problems by updating sample-by-sample.
- Using max-norm regularization compared to the traditional nuclear norm approach for tensor completion.
- Extensive numerical experiments on synthetic data to support the theory of the presented algorithm.

**Audience:**

Yes

**Broader Impact Concerns:**

There are no concerns about the ethical implications of this work that would require adding a Broader Impact Statement.

**Claims And Evidence:**

Yes

**Requested Changes:**

All of the requested changes are critical to securing my recommendation for acceptance.

- Numerical Experiments on Non-synthetic Data:
    - All of the presented experiments were performed on synthetic datasets.  The paper needs results on real-world data, for instance video and hyper spectral data.
    - Additionally, it would be compelling to see how the algorithm performs under different sparsity patterns. The authors only look at random sparsity, and other options like tubal sparsity or structured sparsity (e.g., like that seen in image inpainting) would make OMRTD a more convincing approach.

- Motivation Clarification and Writing Improvement: There were many aspects of the paper that were not motivated clearly, and hence difficult to follow.  Below are a few examples of questions that arose.  There are more, so the reviewer requests a major revision.

    - Why is max-norm regularization helpful? Why does Definition 2 of the tensor max norm require a factorization into $\overline{\mathbf{L}}$ and $\overline{\mathbf{R}}$?
    - What is a sparse noise tensor?  Is it tubally sparse or is it a different notion of sparse noise?
    - What does "(5) can only optimized in a batched manner" mean on Page 4?
    - The convergence analysis in Section 4.3 needs to be fleshed out.  Why are the assumptions made and are they reasonable?  Why do you need to add the additional regularization on $\vec{\mathcal{R}}$ with an $\epsilon$ regularization parameter? etc.
    - The statement in the introduction "Extensive experimental results on the tensor subspace recovery task illustrate that the proposed tensor max-norm **always performs better** than the tensor nuclear norm in terms of convergence rate and robustness." does not seem accurate based on the presented results.  Be sure that your claims are stated appropriately.
    - The statement "Intuitively, the tensor nuclear norm constrains the row norms of $\overline{\mathbf{L}}$ and $\overline{\mathbf{R}}$ on average, while the tensor max-norm constrains the largest $\ell_2$ row norm, hence is **tighter**."  What does "tighter" mean?

- Reproducibility: There is no code provided for reproducibility.  However, this reviewer acknowledges that code may not be able to be provided at this stage of the double blind review process.  This would need to be included in the final version to recommend acceptance.

**Strengths And Weaknesses:**

Strengths:
- The numerical experiments showed the promise of OMRTD for tensor completion, particularly in the cases when there are a greater proportion of missing data.  The algorithm was also shown to be faster than a competing online approach.
- The algorithms were described thoroughly.
- The computationally complexity, particularly in terms of memory allocation, was clearly

Weaknesses:
- Results were only generated for synthetic data and only for tensor completion problems.
- The motivation for the algorithm, the max-norm definition, and more was unclear.
- The writing was hard to follow at times.  For instance, the Convergence paragraph in Section 4.3 explore all of the assumptions and analysis rapidly, making it difficult to verify the accuracy of the statements.

---

> ### Author Response · Authors · 2024-03-23
>
> We thank the reviewer for the suggestions. Actually, our method is not only for tensor completion. The algorithms designed for the complete data and missing data are discussed in Section 4.2 and 4.4, respectively. In the revised manuscript, we had added experiments for background subtraction task using video data. We also polished our paper by addressing the reviewer's comments. Moreover, we will post our code online upon acceptance of this paper. We would like to hear from the review's feedback. Thanks!

---

> > ### Comment · Reviewer_sMav · 2024-04-09
> > **Reviewer comments addressed**
> >
> > The reviewer thanks the authors for their thoughtful and thorough revision.  The authors addressed all of the reviewer's major comments.  The review deems manuscript acceptable for publication.

---

### Review · Reviewer_9zJ8 · 2024-03-07

**Summary Of Contributions:**

This paper proposes the tensor max-norm for online low-rank estimation for tensor data. The formulation of the new max-norm is based on the algebraic framework named tensor Singular Value Decomposition (t-SVD) which finds many applications in visual data modeling in recent years. An algorithm is designed for the proposed online model with analyzed complexity and convergence. Numerical experiments validate the effectiveness of the proposed algorithms.

**Audience:**

Yes

**Claims And Evidence:**

Yes

**Requested Changes:**

1. It is suggested to provide a more detailed and clear analysis of the properties of the proposed max-norm to show its advantages, especially in
- Why do we need to design a new max-norm within the t-SVD framework instead of the Tucker/CP/TT/TN sense?
- What are the advantages of the new max-norm in comparison with other tensor low-rank regularizes like the $*_L$-Spectral $k$-Support norm rather than only the TNN?

2. The proposed online tensor RPCA model seems of few possible applications in today's deep learning age. It is suggested to discuss:
- Are there any possible applications of the proposed online tensor RPCA models to enhance deep learning models?
- What are the possible benefits that t-SVD-based modeling can bring to deep learning? To the best of my knowledge, a typical model is the "stable tensor neural networks" for rapid deep learning which has recently been used for improving robust generalization (e.g., transformed low-rank parameterization can help robust generalization).

3. It is interesting to see some discussions about the theoretical performance of the proposed tensor recovery model, i.e, the estimation error. Anyway, it is not mandatory.

**Strengths And Weaknesses:**

Strengths:

- A new norm: The proposal of tensor max-norm within the t-SVD framework is new to the best of my knowledge.
- Promising numerical experimental results: The experimental results in comparison with (O)TRPCA shows the effectiveness of the proposed algorithms.

Weaknesses:

- Influence: The proposed max-norm-based tensor recovery model seems of limited possible applications in machine learning society.
- Insufficient analysis of the properties of max-norm: The properties of the proposed max-norm are not sufficiently analyzed to support the statement "... and suggest that the tensor max-norm might be a tighter relaxation of the tensor average rank compared to the tensor nuclear norm."
- Reproductivity: The proposed algorithms involve many details, and it may be not easy to implement for readers. The lack of code raises a concern about reproductivity.

---

> ### Author Response · Authors · 2024-03-23
>
> We thank the reviewer for the comments.
>
> 1. Regarding the motivation of the proposed tensor max-norm, as we discussed in the Introduction of the revised manuscript, compared to the Tucker/CP/TT decompositions which consider low-rank structure in the original domain, the t-SVD decomposition takes advantage of structures in the frequency domain. We hope that our proposed max-norm can be a tighter relaxation of the tensor tubal rank in the t-SVD framework than the tensor nuclear norm. One of the advantages of our proposed max-norm is that it can facilitate online learning of low-rank tensors, while it is not clear yet that how to learn low-rank tensors using $\ast_{L}$-Spectral $k$-Support norm in the online settings.
>
> 2. We had applied our method for the online background subtraction task. Unlike deep learning methods, our method does not require training a neural network and it is online by nature. What the max-norm can benefit can bring to deep learning is definitely one of our future works.
>
> 3. One is typically interested in the convergence behavior for online methods. However, the theoretical performance of the proposed tensor recovery model will be considered as future work.

---

### Review · Reviewer_cKGa · 2024-03-12

**Summary Of Contributions:**

This paper contributes a new online algorithm for low-rank tensor factorization under the tensor-tensor product using max-norm regularization of the factors. The authors also propose an extension of their algorithm to data with missing entries. The authors claim the algorithm converges almost surely to a stationary point as t tends to infinity. The method appears to be superior in subspace recovery to the competing OTRPCA method they compare against, lending credence to their claim the tensor max-norm is a tighter approximation to tubal rank, albeit at higher cost in computation time.

**Audience:**

Yes

**Broader Impact Concerns:**

I have no broader impact concerns about this work.

**Claims And Evidence:**

No

**Requested Changes:**

I list my requested changes below by their order as I read them in the paper. For those that are critical to securing my recommendation for acceptance, they are marked with [**CRITICAL**].

1. P3 and formatting throughout: Make sure you use \mathrm{rank} and \mathrm{F} for the Frobenius norm in your LaTeX editor.

P3: Equation (1): I suggest you clarify that \cdot here is ordinary matrix-matrix multiplication. I also suggest you add the dimensions of $\pmb{\mathcal{C}}$, and you that specify how the dimensions of $\pmb{\mathcal{A}}$ and $\pmb{\mathcal{B}}$ must match for the t-product to be valid.

2. P3: Definition 5: The first equality needs to be $\triangleq$.

3. [**CRITICAL**] P4: Section 3: This section overall really could benefit from additional exposition and reorganization to give more intuition for the t-product and t-SVD, particularly for novice readers not familiar with it since t-SVD is still relatively new in the tensor literature. I strongly suggest you discuss how the Fourier domain counterparts of the t-product variables relate to the computation of the t-product in practice. This is a central idea to your work since the tensor max-norm uses this concept too.

4. P4: Section 4: The introductory statement seems to be incomplete? It states "This section first..." but there is no second, third, etc. To my overall point, I think there could be more thoughtful transitions to make the paper flow more nicely. The first sentence of 4.1 reads awkwardly; I would rephrase, and be more precise what you mean by "the basic problem."

5. P4: Section 4.1: I would say "...sparse corruptions, plus Gaussian noise" since your noise model is actually Gaussian from the Frobenius norm in Equation (2), which is to be distinguished from the corruptions that are modeled as sparse.

6. P4: Section 4.1: below Equation (2): I suggest explaining the behavior of $\lambda_1$ and $\lambda_2$.

7. P4: Equation (3): I would make this a formal definition.

8. [**CRITICAL**] P4: Sentence starting "Since the t-product in the spatial domain corresponds..." : To my earlier point, this is really out of context to a reader not already familiar with the t-product since there is no previous discussion about why the Fourier domain variables are important, or the equivalency between the t-product and the matrix-matrix products of the frontal slices in the Fourier domain. I think this needs more supporting exposition, and mention of the product equivalency should be formalized much earlier, such as in Section 3.

9. [**CRITICAL**] P4: "...and hence is tighter to the tubal rank." is what this sentence should read as. How? I don't understand the intuition from this sentence. Is there a proof of this claim?

10. P4: "...we have the following equivalent *optimization problem:* "

11. P4: Last sentence before Prop 1: This sentence is awkward. I would reword and use active voice.

12. [**CRITICAL**] P4: Proposition 1: In what sense are these optimization problems equivalent? Equal objective function values? The same optimal solutions? I think from the proof you mean the former? Can you make this more precise? Just from Prop 1 and the sentence preceding it, it is not immediately clear yet how this allows for sequential optimization, so perhaps think about how this transition can be revised.

13. P5: Algorithm 1: Clarify that $\bar{\mathbf{L}}$ and $\pmb{\mathcal{L}}_t$ are equivalent representations of the basis.

14. P5: First equation: I think there are missing i subscripts on the Z, R, and E variables. This should be congruent with the notation in Equation (7).

14a) [**CRITICAL**] P5: Equation (7): I'm not sure this is the correct optimization problem you are actually solving. I believe you are actually solving the problem (please excuse my own notation here, I will neglect the arrows above the oriented slices...) $$\min_{\pmb{\mathcal{L}}} \sum_{i=1}^N \min_{\pmb{\mathcal{R}}_i, \pmb{\mathcal{E}}_i }\tilde{\ell}(\pmb{\mathcal{Z}}_i, \pmb{\mathcal{L}}, \pmb{\mathcal{R}}_i, \pmb{\mathcal{E}}_i) + ....$$

14b) [Suggestions] Additionally, can you simplify the notation for the frontal faces of the R_i's in the Fourier domain? There is a lot of decoration here that makes it hard to read. Minor suggestion: I would also suggest perhaps using the $\ell_2$ norm on these constraints, since the variables are vectors not matrices. Use \quad to space out the $\forall i$ and $\forall k$.

15. P5: Below Equation (8): I think you are again missing i subscripts in this math. I think the set $\mathcal{P}$ needs to have the dimensions of the variable to specify it is the set of $r \times 1 \times n_3$ slices, and not the full $\pmb{\mathcal{R}}$ tensor.

16. P5: "We assume that each sample is drawn independently and identically distributed from some unknown distribution." Is this true? You are assuming additive Gaussian noise through the Frobenius norm in your objective function.

17. P5: Everything from "This is indeed equivalent to optimizing.... Equation (9)": The organization of this section is a bit odd. Usually in stochastic/online optimization, the exposition begins with introducing the minimization of the expected loss first, and then, as a surrogate, relaxes this to minimization of the empirical loss. This fits better within the context of the online optimization problem to be presented in Section 4.2. Here, this discussion is a bit out of context and does not quite flow well.

18. P6. The tensor inverse under the t-product is undefined as currently in your paper. I recommend including this from the Kilmer & Martin paper and refer to it in the appendix perhaps.

19. P6, Equation (12): One $\eta$ $\forall k$? Or $\eta^{(k)}, \quad k=1,2,3,...$?

20 [**CRITICAL**] P7: Algorithm 3: There is quite a bit of inconsistent and undefined notation in this algorithm block. I will leave it to the authors to find all of the errors, but take care to define all column vectors, and make sure the all \Lbar's are using consistent tensor/matrix notation. You currently index block diagonal matrices with three modes of indices. Also, please specify the convergence criterion.

Does the problem in $\pmb{\mathcal{L}}$ need to be solved exactly every time? If so, what is the convergence criterion? Does your algorithm still work if you solve it inexactly? Can you take just a single subgradient step in $\pmb{\mathcal{L}}$ with a new sample? I understand your convergence results may not hold, but it could significantly speed up your algorithm, since I suspect solving (14) is one of the most expensive steps.

21. P7: First paragraph: $\epsilon$ here is undefined. I think some of these arrow-R's should be indexed by t, as done in Equations (10) and (13).

22. "by optimizing the surrogate function." I know what you mean here, but this is written without any context or explanation why (13) is a surrogate function, i.e., a majorizer. This is a very similar problem to online dictionary learning and matrix factorization, so I *strongly suggest* the authors cite some of the works by Julien Mairal and others.

23. P7: "It is easy to verify that the minimizer of (14)..." do you mean Equation (13)?

24. [**CRITICAL**] P7: Be very careful on this page to define all notational equivalences between the tensor variables and their block-diagonal matrix counterparts in the Fourier domain; some of the equivalent representations for the R variables appear overloaded. Be careful to explain the connection/equivalency between the two optimization problems in Equation (14).

25. P7: You need to include more thoughtful transitions, such as here where you jump to declaring how to compute the subgradient with respect to L without first explaining the optimization approach you will use to solve (14).

25. P8: This should be $\mathbf{U} \triangleq ...$. This is perhaps my personal preference, but using $\mathbf{U}$ to denote a subgradient seems a bit odd...especially since U is used earlier to denote the left singular tensor in the t-SVD. Also, shouldn't $\mathbf{\Theta}$, $\mathbf{Q}$, and the subgradient be indexed by $t$? In the block diagonal representation of $\mathbf{U}$, I suggest being clear that the off-diagonals are zero.

26. P8: Again, make sure all variables denoting slices in time are indexed by $t$.

27. [**CRITICAL**] P8: Convergence: I suggest writing the Convergence subsection more formally. E.g., Assumption 1), Assumption 2), ... Proposition 3), Theorem 4), etc. You say "Following the proof techniques of Shen et al. (2017)...we derive theoretical results..." but then don't provide the formal statements or proofs. It is very hard to verify these claims without explicit proofs.

28. P9: Algorithm 4: specify convergence criterion. Also, why is $\mu$ fixed in the ADMM iterations? Usually this penalty parameter is increased in a geometric progression, such as $\mu^{(t+1)} = \min${$\rho \mu^{(t)}, C$}, where here I use $t$ to denote the ADMM iteration, $\rho > 1$, and $C >> \rho$. This will likely speed up the convergence of your algorithm.

29. [**CRITICAL**] P9: Usually an online algorithm includes a step size/learning rate that varies the tradeoff between newer and older samples, but I don't see this in your work. Can you please comment on why this is, or modify your existing work to account for this? I don't think it would be too hard of an extension to make; the only real changes will probably be in the way the A and B tensors are computed in Equation (14).

30. P9: Right after (16): I think this set should read as $\Omega_t = ${$(i,k) | (i,t,k) \in \Omega$}, since $t$ is fixed.

31. [**CRITICAL**] P9: Equations (17) - (18): Why is the constraint over the entries $\Omega_t$ in (17) dropped in the augmented Lagrangian in (18)? Also, I don't think variable splitting is necessary here. After variable splitting, you still have a least squares problem in D where before you had a similar least squares problem in M with a closed form solution.

32. P10: I think the works OTRPCA (Zhang et al., 2016) and TRPCA (Lu et al., 2020) need to be discussed or at least mentioned in Related Works.

33. P11: No real data experiments? I'm not certain what TMLR's policy is on having real data experiments, but it would certainly strengthen the paper to include real data.

34. P12: "a tensor max-norm based low-rank tensor model" is a little wordy and hard to read.

35. P12: "suggest that the tensor max-norm might be a tighter relaxation" Perhaps use active voice and say, "we conjecture..."

**Strengths And Weaknesses:**

While the algorithm is no doubt interesting and appears to have good performance in practice, there are several weaknesses of this paper that need to be addressed. I detail these more specifically in Requested Changes, but at a high-level:

1. The authors state "we rigorously deduce a new tensor max-norm for tensor decomposition." While the authors define the new tensor-max norm, they do not rigorously prove it is indeed a norm over the space of tensors under the t-product (I suspect this wouldn't be too hard to do...perhaps it is not necessary for a TMLR paper, but to me, the claim of "rigorously deduce" would necessitate this). The authors seem to conjecture throughout the paper that the max-norm is a tighter approximation to tubal-rank, but also claim without proof on P4 "hence it is tighter [sic] to the tubal-rank." I suggest the authors either rigorously provide the theory to their claims or modify their claims appropriately. They also add, "since the tensor max-norm is a more complicated mathematical entity, the development of online methods for the max-norm regularization requires more attention." What do the authors precisely mean by "is a more complicated mathematical entity?" Can you be more specific about the technical challenge this paper addresses?

2. This manuscript needs some reorganization and additional exposition to give more context for important details, particularly surrounding the mechanics of the t-product/t-SVD. Much of the writing in general should be improved to reduce wordiness, awkward phrasing, and passive voice. I suggest the authors include more thoughtful transitions between central ideas.

3. There is often inconsistent, incorrect, or undefined mathematical notation that needs to be resolved.

4. I suggest the theory be made more precise and formal. While I understand theory is optional for a TMLR paper, I think the presentation of Proposition 1 and the convergence guarantees should be cleaned up. The authors claim "we derive several theoretical results" but do not provide formal theorem statements or proofs.

5. I have some concerns about the optimization problems in Equations (7) and (17).

---

> ### Author Response · Authors · 2024-03-23
>
> Thank you very much for these valuable comments! In particular, according to comments 29 and 32, we had modified our algorithm robust OMRTD (Algorithm 3 in the revised manuscript) without variable splitting and $\mu$ is also updated in each iteration, which results in faster convergence (almost 4 times faster)! We had addressed almost all the comments in the revise manuscript. The only thing we would like to highlight here is regarding comment 30. According to Feng et al. (2013) and Shen et al. (2017), $\lambda_1$ and $\lambda_2$ are set only related to the sample dimensions and these parameters are not functions of $t$. As such, because of the $t$'s in the two terms of (14) are canceled out, the learning rate is not decreased when $t$ increases. We hope the reviewer can agree with this.

---

### Decision · Action_Editor_TazL · 2024-04-30

**Recommendation:** Accept with minor revision

**Comment:**

Reviewers were unanimous in their opinion that this was a reasonable and publishable paper. The reviewers requested a lot of changes, and the author made these changes (some of them big, like adding real data to the experiments), and reviewers were happy with this.

I'd like the authors to make a few minor changes (these are either my requests or from the recommendations of the reviewers):
- Section 4, P5: they now say from the Srebro & Shraibman (2005) paper that the matrix max-norm regularizer "could outperform" matrix nuclear norm when the entries are uniformly bounded, but it's not clear what "outperform" here means. Does this mean a tighter approximation to the matrix rank? What other conditions are necessary for it to "outperform" nuclear norm, since "could" does not imply "always"? I think more precision is still needed here.
- Now that you can de-anonymize the manuscript, please include a link to code
- The proof of Theorem 1 is very high-level. For each of the 4 high-level steps, can you be more specific about why this is true, since it can't exactly follow Shen et al for all details. Perhaps justify each step in the appendix
Other little minor things to clean up:
- delete "we understand that" on P7.
- Last sentence of P7: run-on sentence; should read: "bdiag(E). The above..."
- Use a semicolon, not a comma, here on P8: "...sequentially; see details..."

**Audience:**

Tensor factorizations, especially in the online case, are a standard subject that interests many researchers, so Yes, I think there's definitely a sufficient audience.

**Claims And Evidence:**

The paper introduces a new tensor max norm, based on both the matrix max-norm of Srebro et al. and the t-product for tensors of Kilmer and Martin. The paper develops an algorithm to find a factorization based on this criteria and extends it to the tensor-completion setting.

The paper's "claims" include (1) defining the new norm and making some derivations, (2) deriving algorithms and discussing equivalent optimization problems and formulations, and complexity analysis, (3) Theorem 1 on convergence, and (4) numerical experiments.

The "evidence" behind these claims is mostly straightforward (derivations, numerical experiments, etc.).  I found the proof of Theorem 1 a bit high-level and missing some detail.

---

> ### Author Response · Authors · 2024-05-30
> **Camera-ready submission**
>
> Dear Action Editor,
>
> Thank you for soliciting very high-quality reviews for this paper. I have made changes according to your comments and thoroughly proofread the paper. Please let us know if anything further is needed. Thanks!
>
> Best,
> TMLR Paper2152 Authors